# A Method for the Dynamics of Vortices in a Bose-Einstein Condensate: Analytical Equations of the Trajectories of Phase Singularities

Sergi De María-García [1], Albert Ferrando [2], J. Alberto Conejero [1], Pedro Fernández De Córdoba [1] and Miguel Ángel García-March [1,*]

1   Instituto Universitario de Matemática Pura y Aplicada, Universitat Politècnica de València, 46022 València, Spain
2   Department d'Optica, Universitat de València, Dr. Moliner, 50, E-46100 Burjassot (València), Spain
*   Correspondence: garciamarch@mat.upv.es

**Abstract:** We present a method to study the dynamics of a quasi-two dimensional Bose-Einstein condensate which initially contains several vortices at arbitrary locations. The method allows one to find the analytical solution for the dynamics of the Bose-Einstein condensate in a homogeneous medium and in a parabolic trap, for the ideal non-interacting case. Secondly, the method allows one to obtain algebraic equations for the trajectories of the position of phase singularities present in the initial condensate along with time (the vortex lines). With these equations, one can predict quantities of interest, such as the time at which a vortex and an antivortex contained in the initial condensate will merge. For the homogeneous case, this method was introduced in the context of photonics. Here, we adapt it to the context of Bose-Einstein condensates, and we extend it to the trapped case for the first time. Also, we offer numerical simulations in the non-linear case, for repulsive and attractive interactions. We use a numerical split-step simulation of the non-linear Gross-Pitaevskii equation to determine how these trajectories and quantities of interest are changed by the interactions. We illustrate the method with several simple cases of interest, both in the homogeneous and parabolically trapped systems.

**Keywords:** Bose-Einstein condensates; vortices; phase singularities; numerical method

## 1. Introduction

The experiments on vortices in Bose-Einstein condensates (BECs) have a large tradition, starting from landmark experiments where the vortices are created in rotating set-ups [1,2], stirring the condensate [3–5], with phase imprinting [6,7], via obstacles [8] or merging condensates [9]. This research effort has extended for more than twenty years, with recent very interesting experiments showing, e.g., coupling between the atomic spin and orbital-angular momentum [10,11], creation of vortices after free expansion [12], or through a phase transition [13]. In parallel, vortices count with an extensive and strong theoretical literature within the field of BECs, from initial proposals on their possible observation [14,15] and stability [16], through studies based on symmetry on inhomogeneous systems, rings or vortex knots [17–29], to mention just a few examples (for reviews in the topic, see e.g., [30–32]).

Mathematically, a vortex that appears in complex waves is always associated to a phase singularity. A phase singularity occurs in those positions where the intensity of the complex field is zero and the phase is undetermined. Following circuits around these dark spots which are infinitely close to it, the phase increases or diminishes in integer multiples of $2\pi$ (when it increases (diminishes) one says the vortex has a positive (negative) charge). These phase singularities appear in many fields, such as plasma physics [33], fluid

physics [34], atmospheric studies [35] or photonics [36] (with an independent branch of optics called *singular optics* [37–39]).

In this paper, we will use a method initially introduced in the context of photonics [40,41] as a benchmark to study the propagation of initial states containing many vortices with a Gross-Pitaevskii equation. This method has two sides: (1) it allows one to solve a linear Schrödinger equation (a paraxial scalar wave equation in the context of photonics or an ideal Gross-Pitaevskii equation (GPE) with a vanishing coupling constant) when the initial condition contains many phase singularities; (2) it offers naturally equations for the dynamical evolution of the location of each phase singularity, which we call the trajectory of the phase singularity or vortex line. From these equations one can obtain figures of merit of interest, such as merging points for singularities of opposite charge. While in the homogeneous case, the extension from photonics to BECs is merely a pedagogical analogy, in the inhomogeneous (parabolically trapped) case, we generalize here the technique to include potentials with powers of the spatial coordinates. We note that the parabollic case is not common in photonics (except for graded-index optical waveguides [42,43]). Once the technique for the ideal non-interacting case is established, we use the results, both in the homogeneous and trapped cases, to obtain some information insightful to analyze the non-linear cases. To illustrate this, we offer several examples of initial conditions (with one, two and three initial singularities), numerically solving the GPE with a split-step method in the non-linear case, and comparing the results with the ones obtained in the linear case.

The study of the motion of vortices in inhomogeneous (trapped) BECs has an extensive and thorough literature. For a single vortex in a two-dimensional condensate, early numerical studies illustrated that a single (off-axis) vortex precesses around the center of the trap, and the motion was described with effective models from which one can derive a Magnus force, which altogether allowed one to estimate the frequency of the precession [44]. With a variational approach, the effective potential experienced by the vortex was found, proving that indeed the vortex should precess around the cortex core [45], within the regime in which the (inhomogeneous) condensate is large compared with the size of the vortex core (also see Refs. [46,47] for a general study in two and three dimensions and anisotropic harmonic traps, where precession is also discussed, additionally finding, for the case of rotating trap, the angular frequency which stabilizes the vortex). Several works followed, researching diverse aspects of inhomogeneous BECs, generally with a Thomas-Fermi profile, and extending the limits of validity of previous studies or looking to other configurations or systems, e.g., two-component BECs (a non-comprenhesive list includes [48–60]—see also experimental results in Refs. [61–65]). The research effort also focused in the study of the dynamics of two vortices in a BEC, named also vortex dipoles, or more vortices [66–71], researching diverse aspects such as turbulence, chaos, or vortex solitonic structures [72–75] (see also review in [27]). These two or more vortices configurations were studied experimentally in [64,65,76,77]. We highlight here the case of Jones-Roberts solitons, which show elongated elliptical shape, are immune to the snaking instability, and can sustain imprinting of configurations of vortices [78,79]. Also, we note here that a Magnus force also appeared in some of the co-authors previous work, where it was considered a highly charged vortex located on axis and broken by a sudden turn on of an optical lattice (with a squared symmetry)—see [80].

We emphasize here that the present work diverts generally in one fundamental aspect from this list of works. The initial condition is not a vortex embedded in a BEC with Thomas-Fermi profile or a Jones-Roberts soliton with a phase pattern imprinted. In the examples used to illustrate the technique introduced here we consider one, two or three vortices inside a disc of density (see Figure 1). Generally, the technique is valid for an arbitrary number of vortices at arbitrary positions, see Equation (4). In all initial conditions, the vortices are located within the ring of density, and the dynamics of the singularities occurs mostly there. These initial conditions are more natural for interactions which are attractive, but here we also consider its dynamics in the repulsive case. As discussed in conclusions and outlook this paper has to be considered as a first step of a research program

which will extend the method to other initial conditions and compare with the effective models developed in the literature. The paper is organized as follows: In Section 2, we introduce the method both for the homogeneous and trapped (linear) cases. In Section 3, we exemplify the method in the homogeneous case with some linear and non-linear examples (attractive and repulsive interactions, two and three initial singularities), and compare both. In Section 4, we undertake a similar analysis for the trapped case (here with one, two and three initial singularities). We end the paper with some conclusions and outlook in Section 5.

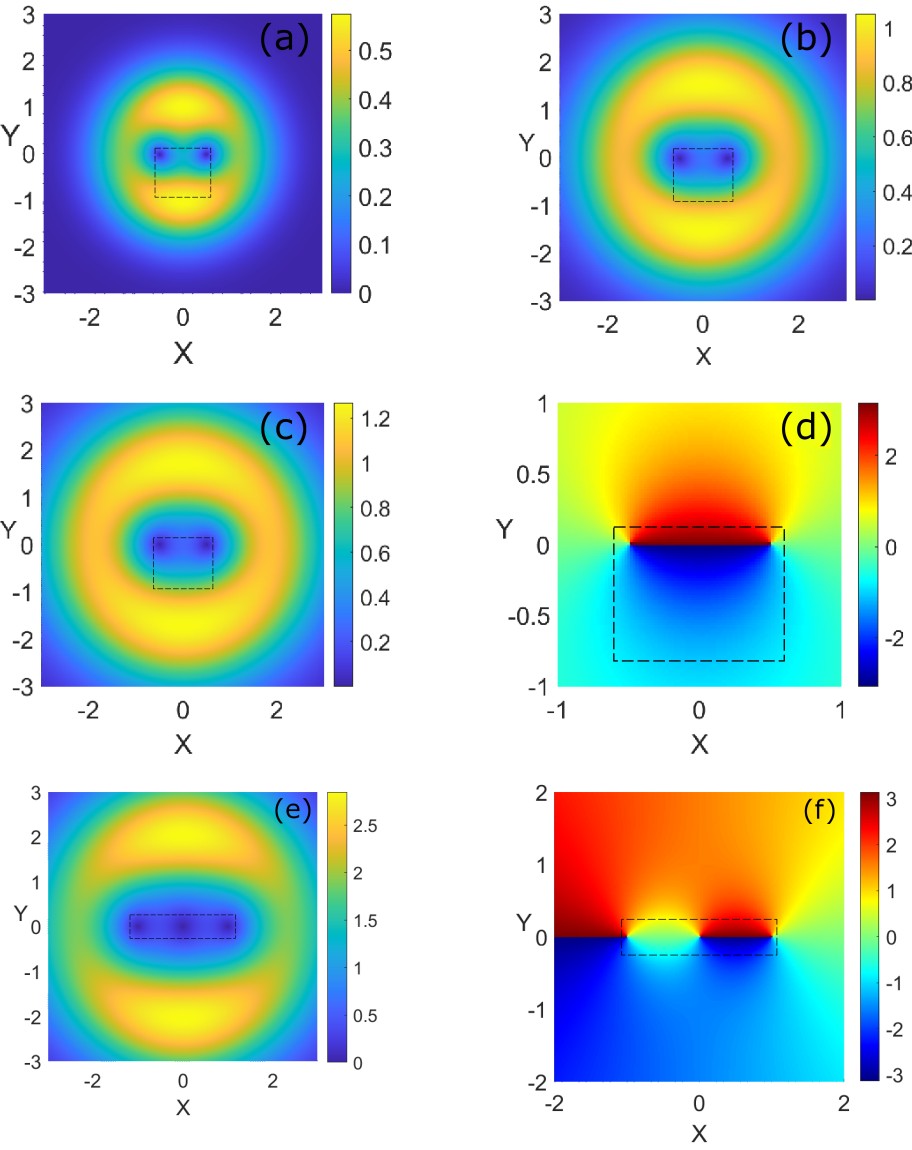

**Figure 1.** (**a**–**c**) Amplitude (square root of density) of the initial condition for $N = 5, 20$ and $30$ respectively, when there are initially two vortices of opposite charge ($q = \pm 1$), located at $(\pm 0.5, 0)$. (**d**) corresponding phase. The phase is the same in the three cases because, due to the form of Equation (4), one decides the position of the singularities with the coordinates of the initial vortices, and therefore the phase profile. (**e**,**f**) amplitude and phase, respectively, for three vortices, one negatively charged ($q = -1$) in the origin and two, positively charged ($q = 1$) located at $(\pm 1, 0)$. In all plots, the dashed black square is the window represented in other figures in the paper.. Also notice that the computational box is much larger than the one represented in all figures.

## 2. Model and System

The dynamics of a system of ultracold Bose-Einstein condensed bosons is governed by the Gross-Pitaevskii equation (GPE)

$$- i\hbar \frac{\partial \phi(\mathbf{r}, t)}{\partial t} = \left[ -\frac{\hbar^2}{2m} \nabla_\perp^2 + V(\mathbf{r}) + g|\phi(\mathbf{r}, t)|^2 \right] \phi(\mathbf{r}, t), \tag{1}$$

with $\mathbf{r} = (x, y)$, $\nabla^2$ the Laplacian, $m$ the mass of the bosons, $V(x, y)$ the external potential, and $g$ the coupling constant. We assume a 2D system characterized by the complex wave function $\phi(\mathbf{r}, t)$ of the condensed bosons, with dynamics frozen in the third direction $z$ due to tight confinement, giving rise then to a quasi-two-dimensional Bose-Einstein condensate. In the following, we will use complex position operators, $\hat{w}$ and $\hat{\bar{w}}$, which we define as $\hat{w} = \hat{x} + i\hat{y}$ and $\hat{\bar{w}} = \hat{x} - i\hat{y}$. Their associated momenta are $\hat{p} = -i\partial/\partial \hat{w}$ and $\hat{\bar{p}} = -i\partial/\partial \hat{\bar{w}}$. The commutations relations are

$$[\hat{w}, \hat{p}] = \hat{w}\hat{p} - \hat{p}\hat{w} = -i\,\hat{w}\overbrace{\frac{\partial}{\partial w}}^{0} + i\frac{\partial}{\partial w}\hat{w} = i,$$

$$[\hat{\bar{w}}, \hat{\bar{p}}] = \hat{\bar{w}}\hat{\bar{p}} - \hat{\bar{p}}\hat{\bar{w}} = -i\,\hat{\bar{w}}\overbrace{\frac{\partial}{\partial \bar{w}}}^{0} + i\frac{\partial}{\partial \bar{w}}\hat{\bar{w}} = i. \tag{2}$$

From here on we omit the hats in the operators. These definitions allow one to write the GPE as

$$- i\hbar \frac{\partial \phi(w, \bar{w}, t)}{\partial t} =$$
$$\left[ -\frac{\hbar^2}{2m} \nabla_w^2 + V(w, \bar{w}) + g|\phi(w, \bar{w}, t)|^2 \right] \phi(w, \bar{w}, t). \tag{3}$$

Now the Laplacian is, explicitly, $\nabla_w^2 = \frac{\partial^2}{\partial w^2} + \frac{\partial^2}{\partial \bar{w}^2}$. We will consider initial conditions of the form

$$\phi(w, \bar{w}, 0) = \prod_{i=1}^{N_A} (w - \mathbf{a}_i) \prod_{j=1}^{N_B} (\bar{w} - \mathbf{b}_j) \phi_{00}(w, \bar{w}), \tag{4}$$

with

$$\phi_{00}(w, \bar{w}) = A \exp\left[ -|w|^2/2\sigma^2 \right], \tag{5}$$

which is a Gaussian of amplitude determined by $A$ and width $\sigma$ (see some examples of initial conditions described by this equation in Figure 1). We note that in the case of BECs the amplitude to the square is the density of the condensate. We normalize the initial condition (4) to the total number of atoms, $N$. We keep $N$ in the initial condition normalization to show explicitly how it appears in a quantity of interest like the merging point in the non-interacting case and because it determines the distance of the ring of density to the positions of the singularities (because we fix the position of singularities—see the examples for two singularities in Figure 1). With this form, the initial condition (4) is a combination of singularities embedded in a Gaussian wave function. Here, there are $N_A$ ($N_B$) singularities of charge $+1$ ($-1$) located at positions $\mathbf{a}_i = (w_i, \bar{w}_i)$ ($\mathbf{b}_j = (w_j, \bar{w}_j)$). The topological charge $q$ of one singularity is defined as the circulation of the gradient of the phase of the complex field $\phi(w, \bar{w}) = |\phi(w, \bar{w})|e^{i\theta(w, \bar{w})}$ in circuits around the positions of the phase singularities $\mathbf{a}_i$ ($\mathbf{b}_j$), and very close to them, divided by $2\pi$, i.e.,

$$q = \frac{1}{2\pi} \oint_C \nabla\theta(s)ds. \tag{6}$$

In plane words, the phase grows (decreases) $2\pi$ along a circuit around $\mathbf{a}_i$ ($\mathbf{b}_j$) assuming this circuit does not encircle any other singularity. For simplicity of notation we consider in this point initial conditions which include only individual singularities of charge $q = \pm 1$ (as in the initial condition (4)). To consider an initial condition embedding individual singularities with larger (in modulus) charge is possible: by including powers of factors, i.e., $(w - \mathbf{a_i})^{\mathbf{k}}$ or $(\bar{w} - \mathbf{b_j})^{\mathbf{s}}$, with $k$ and $s$ integers, we would consider singularities of charge $k$ or $-s$. We note that the total winding number of the initial condition is the sum of all topological charges, as it has to be calculated with the same formula but in a circle that surrounds all singularities inside. We refer to [81] for a pedagogical discussion on symmetry, angular momentum, topological charge and winding number, with the same definitions that we use in this manuscript.

Let us now discuss a method to easily find the analytical solutions of the GPE (1) for multisingular initial conditions, Equation (4), in the linear case, $g = 0$. This method was introduced in the context of the paraxial wave equation in optics [40,41] (see also [82] for an application to a different system). In the following sections we will compare the solutions analytically obtained for the linear case, with some interesting examples calculated numerically for the full non-linear equation.

The first step is to expand the initial condition (4) as powers of $w$ and $\bar{w}$ as follows

$$\phi(w, \bar{w}, 0) = \sum_{\{n, \bar{n}\}} t_{n, \bar{n}} w^n \bar{w}^{\bar{n}} \phi_{00}(w, \bar{w}), \tag{7}$$

where $\{n, \bar{n}\}$ is the set which includes all powers of the form $w^n \bar{w}^{\bar{n}}$ after the expansion and $t_{n, \bar{n}}$ are complex coefficients. Let us define the functions

$$\phi_{n, \bar{n}}(w, \bar{w}) = w^n \bar{w}^{\bar{n}} \phi_{00}(w, \bar{w}), \tag{8}$$

which are defined by means of two quantum numbers, $n$ and $\bar{n}$.

To give a meaning of these quantum numbers, let us see that they relate to the angular momentum and radial nodes quantum numbers. We define

$$\ell = n - \bar{n} \quad \text{and} \quad p = \min(n, \bar{n}). \tag{9}$$

Let us first consider that $n \geq \bar{n}$. Then, one can write Equation (8) as

$$\phi_{\ell, p}(w, \bar{w}) = |w|^{2p} w^\ell \exp\left[-|w|^2 / 2\sigma^2\right], \tag{10}$$

which, in polar coordinates, is

$$\phi_{\ell, p}(r, \theta) = r^{2p} r^\ell \exp[i\ell\theta] \exp\left[-r^2 / 2\sigma^2\right]. \tag{11}$$

This is a scattering mode (as termed in [41]) labelled by two quantum numbers: the angular momentum quantum number $\ell = n - \bar{n} > 0$ and the radial quantum number $p = \bar{n}$. For the case $n < \bar{n}$ one obtains that $p = n$ and $\ell = n - \bar{n} < 0$, and one can write Equation (8) as

$$\phi_{\ell, p}(w, \bar{w}) = |w|^{2p} \bar{w}^{|\ell|} \exp\left[-|w|^2 / 2\sigma^2\right], \tag{12}$$

which, in polar coordinates, is

$$\phi_{\ell, p}(r, \theta) = r^{2p} r^{|\ell|} \exp[i\ell\theta] \exp\left[-r^2 / 2\sigma^2\right], \tag{13}$$

which is again a scattering mode. Then, we can expand the initial condition in terms of the scattering modes (which form a basis) as

$$\phi(r, \theta, 0) = \sum_{\{\ell, p\}} t_{\ell, p} \phi_{\ell, p}(r, \theta), \tag{14}$$

that is, as a linear combination of scattering modes with a vortex of charge $\ell$ at the origin. This function (which contains the same information as Equations (4) and (7)) we normalize to the number of atoms $N$.

As a second step, let us show how to find the analytical expression for the evolution of the functions in this basis. First we notice that, in the complex variables, we can write the effective Hamiltonian leading to Equation (1) with $g = 0$ as $H = p\bar{p} + V(w, \bar{w})$. We will consider two cases: (i) the homogeneous case $V(w, \bar{w}) = 0$; and (ii) that the atoms are trapped in a parabolic potential, $V(w, \bar{w}) = \frac{1}{2}m\omega^2|w|^2$ (in cartesian coordinates, $V(x, y) = \frac{1}{2}m\omega^2(x^2 + y^2)$). The first case corresponds to the optical case discussed in [40,41], where the technique used in this paper was introduced for photonic systems. Here we generalize this technique to the trapped case and compare with the non-linear case, which is the most relevant for ultracold atom systems.

The evolution operator is $U(t) = \exp[itH/\hbar]$. Using the commutation relations (2), one has that $[w, p\bar{p}] = i\bar{p}$ and $[\bar{w}, p\bar{p}] = ip$. Also, we use that $[w, w\bar{w}] = 0$ and $[\bar{w}, w\bar{w}] = 0$. Then, we obtain (We notice the known property that if $[A, B] = k$, then $[A, \exp(\lambda B)] = \lambda k \exp(\lambda B)$)

$$[w, U(t)] = -\bar{p}\frac{t}{\hbar}U(t),$$
$$[\bar{w}, U(t)] = -p\frac{t}{\hbar}U(t). \tag{15}$$

The last step is to use this commutators to perform evolution. Consider an initial condition which is a scattering mode $\phi_{\ell,p}(w, \bar{w}, 0)$. To evolve it to time $t$ we apply the evolution operator, $\phi_{\ell,p}(w, \bar{w}, t) = U(t)\phi_{\ell,p}(w, \bar{w}, 0)$. We now use that the scattering mode can be written as $\phi_{\ell,p} = w^{p+\ell}\bar{w}^p\phi_{0,0}$ if $\ell \geq 0$ or $\phi_{\ell,p} = \bar{w}^{p+|\ell|}w^p\phi_{0,0}$ if $\ell < 0$ (see Equations (10) and (12)). Then, using the commutation relations (15) we obtain:

$$
\begin{aligned}
&\phi_{\ell,p}(w, \bar{w}, t) \\
&= U(t)\phi_{\ell,p}(w, \bar{w}, 0) = U(t)w^{p+\ell}\bar{w}^p\phi_{0,0}(w, \bar{w}, 0) \\
&= \left(w + \bar{p}\frac{t}{\hbar}\right)^{p+\ell}\left(\bar{w} - p\frac{t}{\hbar}\right)^p U(t)\phi_{0,0}(w, \bar{w}, 0),
\end{aligned} \tag{16}
$$

if $\ell \geq 0$ and if $\ell < 0$

$$
\begin{aligned}
&\phi_{\ell,p}(w, \bar{w}, t) \\
&= U(t)\phi_{\ell,p}(w, \bar{w}, 0) = U(t)\bar{w}^{p+|\ell|}w^p\phi_{0,0}(w, \bar{w}, 0) \\
&= \left(\bar{w} + p\frac{t}{\hbar}\right)^{p+|\ell|}\left(w - \bar{p}\frac{t}{\hbar}\right)^p U(t)\phi_{0,0}(w, \bar{w}, 0).
\end{aligned} \tag{17}
$$

In Equations (16) and (17) one can obtain analytically the evolved scattering mode $\phi_{\ell,p}(w, \bar{w}, t)$ if one can evolve the function $\phi_{0,0}(w, \bar{w}, 0)$. That is if one can obtain the function

$$G(w, \bar{w}, t) = U(t)\phi_{0,0}(w, \bar{w}, 0). \tag{18}$$

We call this function the *generating function* of the dynamics. For the homogeneous case, $V(w, \bar{w}) = 0$, with an initial condition as (4) with $\psi_{0,0}$ in (5), the generating function is conventionally found in simple systems (like free particle in Schrödinger equation).

For the inhomogeneous case, $V(w, \bar{w}) \neq 0$, with the same initial condition, and for the particular case where the potential is parabolic, $V(x, y) = \frac{1}{2}m\omega^2(x^2 + y^2)$ that is

$V(w, \bar{w}) = \frac{1}{2} m \omega^2 |w|^2$, one can also perform the evolution analytically, provided $g = 0$. We will use the two-dimensional harmonic oscillator propagator (see e.g., [80])

$$
\phi_{00}(x, y, t) = \frac{\csc(t)}{2\pi i} * \exp\left[\frac{-i(x^2 + y^2)\cot(t)}{2}\right] \cdot
$$
$$
\iint_{\mathbb{R}^2} \exp\left[-i\left(\frac{(x_0^2 + y_0^2)\cot(t)}{2} + (x_0\, x + y_0\, y)\csc(t)\right)\right] \cdot \tag{19}
$$
$$
\phi_{00}(x_0 + i\, y_0, x_0 - i y_0, 0) dx_0 dy_0,
$$

which we have written for time adimensionalized by the frequency $\omega$ of the harmonic potential and for space adimensionalized with the harmonic oscillator length $a_{\mathrm{ho}} = \sqrt{\hbar/m\omega}$. Then, we can calculate the explicit expression for the generating function $G(x, y, t)$ in cartesian coordinates

$$
G(x, y, t) = \frac{-A\sigma^2}{s'(t)} \exp\left[\frac{-(x^2 + y^2)s(t)}{2i * s'(t)}\right], \tag{20}
$$

where $s(t) = \sigma^2 \sin(t) + 2i\pi \cos(t)$. We note that this last expression is $2\pi$-periodic.

Now, to obtain any evolved scattering mode with Equations (16) and (17) one has to apply several operations to the generating function. To write this in a succinct manner let us define the raising and lowering operators:

$$
\ell_+(t) = w - i\frac{t}{\hbar}\frac{\partial}{\partial \bar{w}}, \quad \text{and} \quad \ell_-(t) = \bar{w} - i\frac{t}{\hbar}\frac{\partial}{\partial w}. \tag{21}
$$

As shown in [41], these operators actually increase/decrease the angular momentum operator of a general scattering mode $\phi_{\ell, p}$ by one unit. Also let us define the operator $\Delta = \ell_+ \ell_-$ which was also shown in [41] to increase its radial quantum number by one unit. Then, the evolution of a scattering mode can be written as follows:

$$
\phi_{\ell, p}(w, \bar{w}, t) = \ell_{\mathrm{sign}(\ell)}^{|\ell|}(t)\Delta^p(t)U(t)\phi_{0,0}(w, \bar{w}, 0)
$$
$$
= \ell_{\mathrm{sign}(\ell)}^{|\ell|}(t)\Delta^p(t)G(w, \bar{w}, t). \tag{22}
$$

Let us summarize the method to solve the GPE, Equation (1) when $g = 0$ and the initial condition is a multisingular field of the form Equation (4). The method is as follows:

1. Expand the initial condition Equation (4), as a linear combination of terms $w^n \bar{w}^{\bar{n}}$ of the form (7).
2. Use definition (9) to write this expansion as an expansion in terms of the scattering modes, Equation (14).
3. Evolve each scattering mode. To this end, find the generating function (which is to evolve the fundamental mode $\Phi_{0,0}$ in this case). Then use Equation (22) to find each evolved scattering mode.
4. Use the evolved scattering mode to find the solution, via Equation (14), that is,

$$
\phi(r, \theta, t) = \sum_{\{\ell, p\}} t_{\ell, p}\phi_{\ell, p}(r, \theta, t). \tag{23}
$$

Step 3 involves several repetitive operations on the generating function. This can be done analytically in the homogeneous case, $V(w, \bar{w}) = 0$. The evolved scattering modes can be obtained systematically in this case, using the $F$-polynomials introduced in [41]. These give closed expressions for the evolved scattering modes. We remark here that the $F$-polynomials, which are tabulated in [41], obey recurrence relations which facilitate its computation. We also note that, in the inhomogeneous case, if the potential involves any combination of $w$ and $\bar{w}$ and its powers, obtaining the generating function cannot be,

in general, performed analytically. However, one can still use the method evolving $\phi_{0,0}$ numerically and then, to evolve the scattering modes, perform the operations included in Equation (22) again numerically. Finally, one builds the solution $\phi(r, \theta, t)$ as in step 4 (Equation (23)).

We finally mention that the second side of this method, both for the homogeneous and inhomogeneus cases in the non-interacting $g = 0$ case, is that it permits one to obtain algebraic equations for the singularity trajectories. These in turn allow one to determine several properties related to the position of the phase singularities of the evolved solution via solving these algebraic equations. To illustrate the method, we offer several examples in the next section. Also, we compare our results with the non-linear case. To this end, we use the split-step method to solve the Equation (1) in the non-linear ($g \neq 0$) homogeneous and inhomogeneous cases.

## 3. Some Examples in the Homogenous System

### 3.1. Two Initial Singularities, One Positive and One Negative

First, we will discuss the case with a field with a positive singularity in $\mathbf{a} = (a, 0)$, and a negative one in $\mathbf{b} = (b, 0)$

$$\phi(w, \bar{w}, 0) = (w - a)(\bar{w} - b)\phi_{00}(w, \bar{w}). \tag{24}$$

We note that the same method can be used for singularities at arbitrary initial conditions, but the results obtained are long and we omit them here for the sake of simplicity. We remark here that in all calculations below, we use the adimensionalization mentioned above for the trapped case (see paragraph after Equation (19)), that is time is scaled by a trapping frequency $\omega$ and positions by the related harmonic oscillator length $a_{\mathrm{ho}} \equiv \sqrt{\hbar/m\omega}$, with a generic mass and generic frequency. Then, by choosing mass and frequency, one can recover units.

We first expand the initial condition (24) in powers of $w$ and $\bar{w}$

$$\phi(w, \bar{w}, 0) = (|w| - a\bar{w} - bw + ab)\phi_{00}(w, \bar{w}). \tag{25}$$

Following step 2 we produce

$$\begin{aligned} \phi(w, \bar{w}, t) =& \phi_{01}(w, \bar{w}, t) - a\phi_{-10}(w, \bar{w}, t) \\ &- b\phi_{10}(w, \bar{w}, t) + ab\phi_{00}(w, \bar{w}, t). \end{aligned} \tag{26}$$

And now, step 3 is accomplished using Equation (22), obtaining

$$\begin{aligned} \phi_{01}(w, \bar{w}, t) =& \left(\frac{i\sigma^2}{q(t)}\right)\left(\frac{t\sigma}{\pi q(t)}\right)(1 - \gamma(t)|w|^2)\exp\left[\frac{-i\pi|w|^2}{q(t)}\right] \\ \phi_{10}(w, \bar{w}, t) =& \, w\left(\frac{i\sigma^2}{q(t)}\right)^2 \exp\left[-\frac{i\pi|w|^2}{q(t)}\right] \\ \phi_{-10}(w, \bar{w}, t) =& \, \bar{w}\left(\frac{i\sigma^2}{q(t)}\right)^2 \exp\left[-\frac{i\pi|w|^2}{q(t)}\right] \\ \phi_{00}(w, \bar{w}, t) =& \left(\frac{i\sigma^2}{q(t)}\right)\exp\left[-\frac{i\pi|w|^2}{q(t)}\right], \end{aligned} \tag{27}$$

where $q(t) = t + i\sigma^2$ and $\gamma(t) = \dfrac{\pi\sigma^2}{tq(t)}$. Now the solution can be built from Equation (25), realizing step 4,

$$\phi(w,\bar{w},t) = \left(\frac{i\sigma^2}{q(t)}\right)\exp\left[-\frac{i\pi|w|^2}{q(t)}\right]\cdot$$
$$\left[\left(\frac{t\sigma}{\pi q(t)}\right)(1-\gamma(t)|w|^2)-a\bar{w}\left(\frac{i\sigma^2}{q(t)}\right)-bw\left(\frac{i\sigma^2}{q(t)}\right)+ab\right]. \tag{28}$$

Equation (28) is the solution of the evolution for all $t$, given $g = 0$. We compared this solution with those obtained with a numerical simulations of the GPE (1) for $g = 0$ obtaining complete agreement. The numerical simulations were performed with a split-step method, which is a spectral method used conventionally when solving non-linear Schrödinger equations, appearing in various fields, like non-linear optics (see e.g., [83], where code can be found in appendices). This method is valid for $g = 0$, for $g > 0$ (repulsive case) and $g < 0$ (attractive case). For all simulations we use a computational domain which is a box where $x, y \in [-10, 10]$ in adimensional units. We use a discretization with $M = 1024$ points in each side (and then $\Delta x = \Delta y \approx 0.02$ a.u. For the time step we use also $\Delta t = 0.02$ a.u. We calculate for very long times which depend on the simulations. That is, the simulation time is limited for the repulsive and attractive case when the density reaches the boundaries, and starts to interfere with the central dynamics. This maximum computational time is almost in all (homogeneous) cases $t_{\max} = 15$ a.u. In the attractive case, in some instances the computational time is shorter than that because simulation stops when instability occurs. For the trapped case, we can perform much longer simulations. For every example shown in this paper, we also calculated with the split-step method in the non-interacting case to check that the results coincide with the analytical solution. We checked in all calculations that the energy is conserved during the whole simulation.

Once the linear solution is established, it is interesting to compare with the non-linear cases. We show a typical non-linear evolution in Figure 2, where we plot the amplitude and phase after some time evolution for the repulsive case, with $N = 30$ and $g = 0.4$. We present two exemplary times, at $t = 1.5$ a.u. and $t = 2.5$ a.u., which are a bit before and a bit after the merging time (the time at which the two singularities have annihilated each other). After merging time there is no singularity present in the phase profile. In Figure 3, we plot the same but for larger interactions strength, $g = 0.6$. Now for the same time as in previous example, $t = 2.5$ a.u., merging of the two singularities has not occurred yet.

For the attractive case, we plot in Figure 4 the amplitude and phase after some time evolution of different initial conditions, that is, for $t = 2.2$ a.u., $g = -0.5$ and $N = 5$, $N = 20$ and $N = 30$. First we see that the dynamics of the amplitude is very different in this case when compared to the repulsive one, as expected. In the repulsive one the amplitude spreads and widens, while in the attractive case it focuses more and more, leading eventually to instability. The case $N = 30$ shows an example just a bit before and after instability occurs (see panels (e) to (h)). Also, in view of Figure 4 we see that for this $g$, merging time is larger than $t = 2.2$ a.u. for $N = 5$ and shorter for $N = 20$ (there is no singularity in the profile for $N = 20$ yet there are two singularities for $N = 5$). For $N = 30$ there are two singularities because $N$ has the effect to make the merging to take place at larger $|g|$, for large enough $N$ (that is for this $g$ merging time is larger for $N = 30$). In summary, the behavior is very different for different $N$: whilst for $N = 5$ and $N = 30$ we see the singularities has not merged yet, for $N = 20$ merging has occurred. Also for $N = 30$ we see an example just before instability takes place.

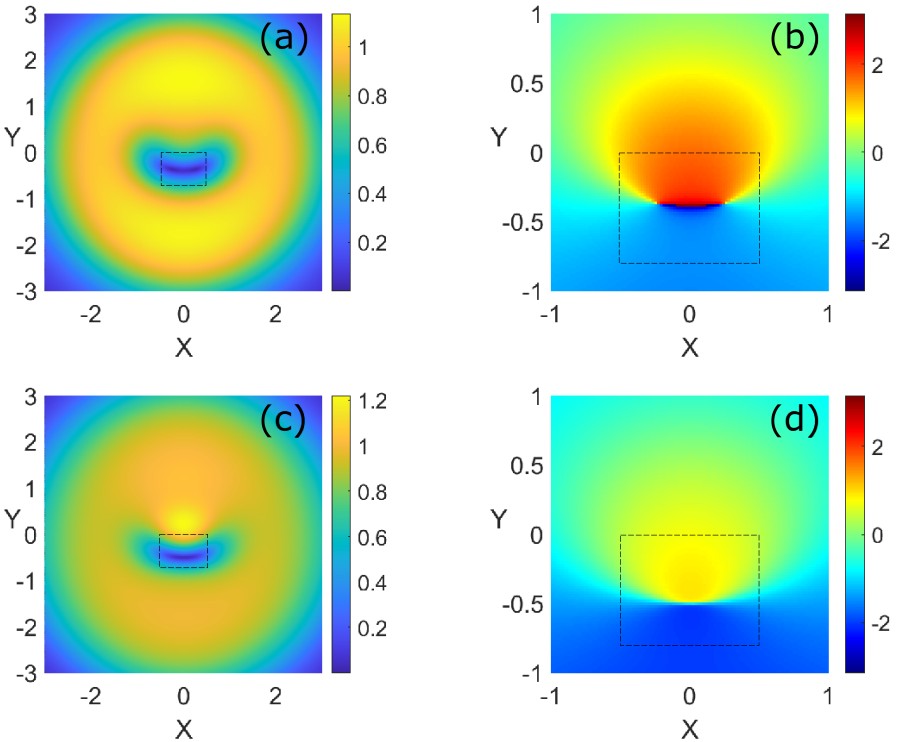

**Figure 2.** For $N = 30$ amplitude and phase for $g = 0.4$ (repulsive case) after $t = 1.5$ a.u. (**a**,**b**) and after $t = 2.5$ a.u. (**c**,**d**). Notice we plot a box which is smaller than the computational box. For $t = 1.5$ a.u. merging has not yet occurred but for $t = 2.5$ a.u. the two singularities have merged leaving a phase profile without singularities. The dashed black squares mark the plotting box in other figures below.

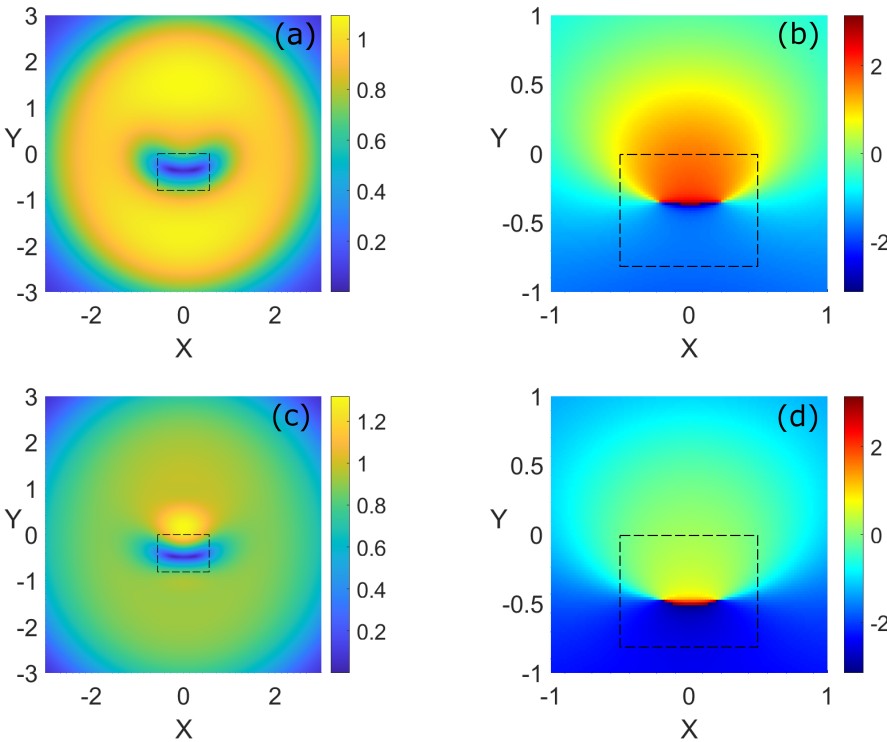

**Figure 3.** For N = 30 amplitude and phase for g = 0.6 (repulsive case) after t = 1.5 a.u. (**a**,**b**) and after t = 2.5 a.u. (**c**,**d**). Here, merging has not occurred at $t = 1.5$ a.u. nor at $t = 2.5$ a.u., contrarily as in the case with $g = 0.4$. The dashed black squares mark the plotting box in other figures below.

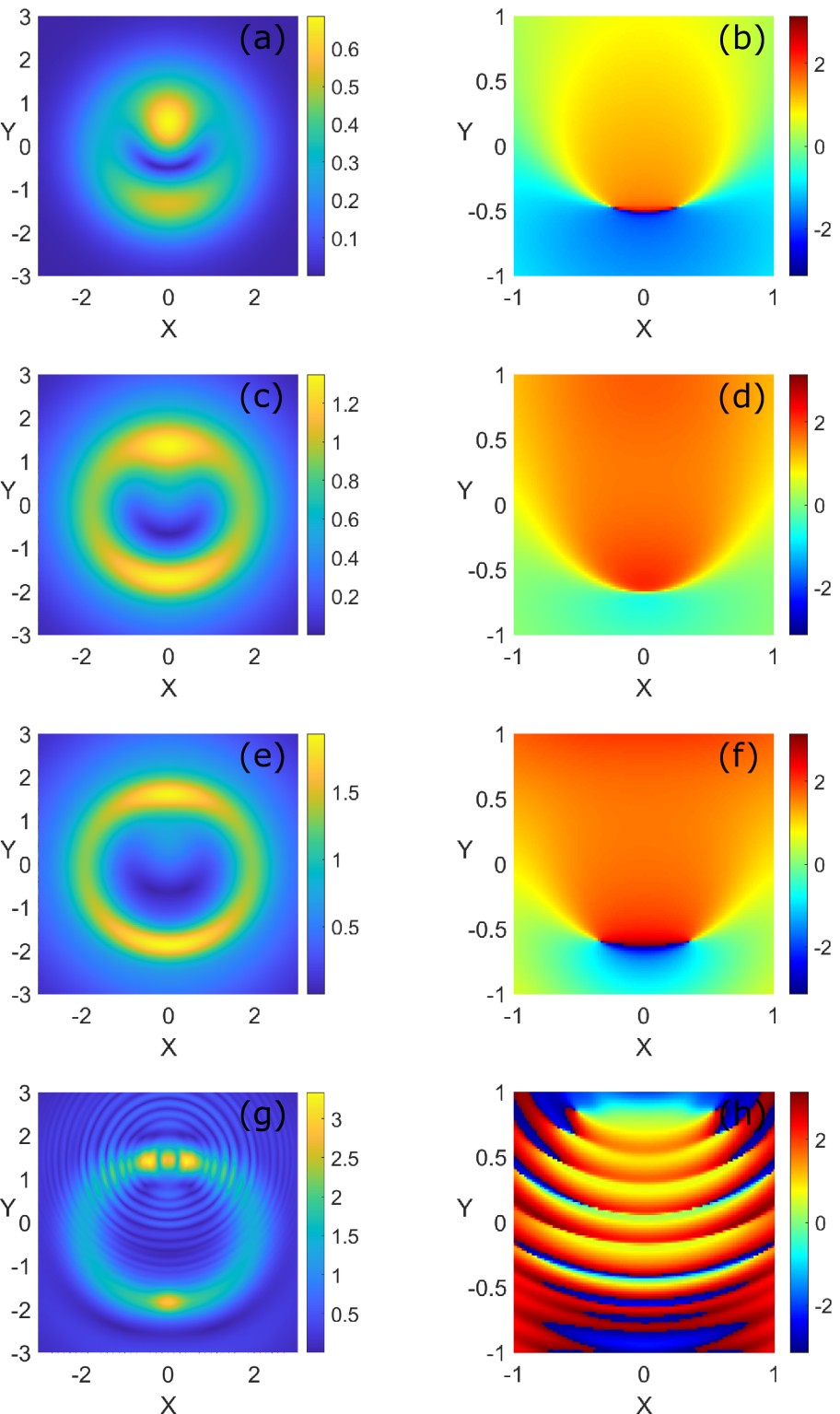

**Figure 4.** Amplitude and phase for $g = -0.5$ (attractive case) after $t = 2.2$ a.u. for $N = 5$ (**a,b**), $N = 20$ (**c,d**), and $N = 30$, (**e,f**). For $t = 2.2$ a.u. and $N = 5$ and $N = 30$ merging has not yet occurred. However, for $N = 20$ it has taken place (see Figure 4 for the cases of $N = 5$ and $N = 30$. We also plot in panels (**g,h**) the amplitude and phase for $N = 30$ and $t = 4.0$ a.u., a time in which instability has occurred and the simulation is not valid anymore.

We see here that the trajectories followed by the singularities, alternatively referred to as vortex lines, can show intricate behavior. It is helpful to determine these trajectories for the linear case and the compare with the non-linear cases. In these non-linear cases, as we will see, the trajectories change due to the effect of the density in the dynamics. It would be very interesting to study, in the context of the literature mentioned in the last paragraph of the introduction, the behavior in the non-linear case, and whether this method can shed some light in this direction. However, here we focus on introducing the method to find equations for the vortex lines in the non-interacting case, and use it to compare with the interacting case.

To obtain equations of the vortex lines we note that Equation (28) gives more information. Equating to zero this solution we can find in which points the solution is zero, for all $t$. This will provide equations for the trajectories of the singularities. For this particular case, the equation is

$$t\sigma^2(1 - \gamma(t)|w|^2) - a\bar{w}i\pi\sigma^2 - bwi\pi\sigma^2 + ab\pi q(t) = 0, \tag{29}$$

which we solve together with its complex conjugate. From here we can find some figures of merit. In the case of two singularities, we expect that at certain merging time $t_{\mathrm{m}}$, the two singularities converge and annihilate each other. This merging time can be found analytically from Equation (29). For $b = -a$ it is

$$t_{\mathrm{m}} = \frac{2a^2\pi\sigma}{\sqrt{-4a^2\pi + \sigma^2}}. \tag{30}$$

Notice that this expression depends on $\sigma$ and $a$. Given $a$, for different numbers of atoms, changing $\sigma$ one can implement the normalization of the initial condition, that is, the number of atoms $N$. In Figure 5 we plot how the merging time changes with the number of atoms in the initial condition, $N$. As shown, it is a decreasing function of $N$. Notice that, due to the way we wrote the initial condition, Equation (4), $N$ governs the distance between the singularities and the ring of density surrounding them, and the shape of the density. The dependence of the initial conditions density shape with $N$ is illustrated in Figure 1.

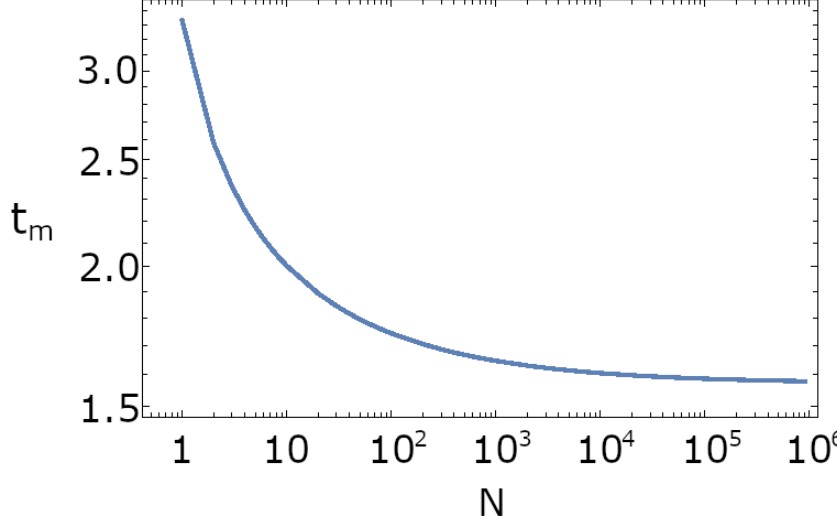

**Figure 5.** Decrease of merging time with number of atoms $N$ in the non-interacting case.

In Figure 6a we plot the position of the two singularities as a function of time (the trajectories or vortex lines), for $a = 0.5$, $b = -0.5$, and $N = 30$ atoms (which gives $\sigma^2 = 10$), from the analytical solution, Equation (23). We notice that to find the position of the singularities requires a dedicated numerical method. For a large enough number of singularities it may require sophisticated methods, see, e.g., Refs. [84,85] but in our case

we used a simple method. The analytically calculated merging time is $t_m = 1.9$ a.u. In Figure 6b–d we plot the position of the two singularities as a function of time for repulsive interactions, when $g = 0.4, 0.54$ and $g = 1$, with $N = 30$.

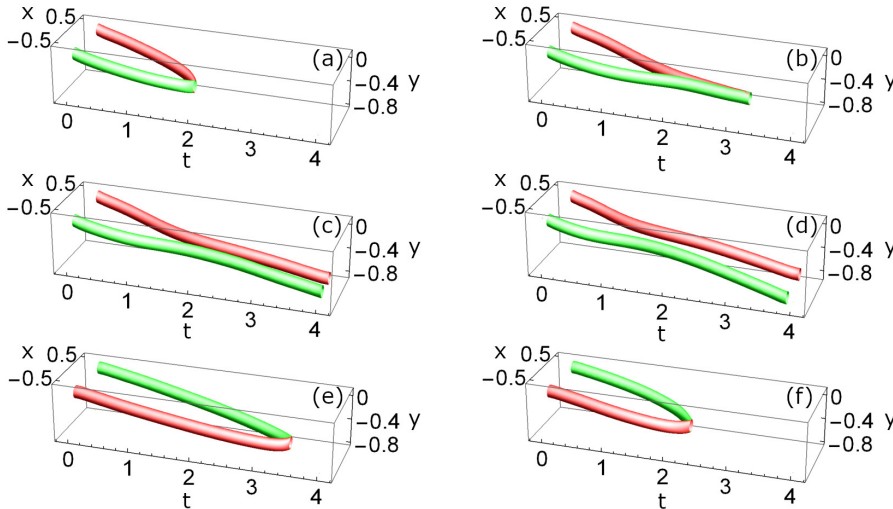

**Figure 6.** (**a**) Trajectories of phase singularities when the initial condition contains a phase singularity of charge $q = 1$ at **a** $= (0.5, 0)$ and another phase singularity of charge $q = -1$ at **a** $= (-0.5, 0)$, when $g = 0$ and $N = 30$. The singularities annihilate at $t_m = 1.9 \, a.u.$ (**b**–**d**) Trajectories of phase singularities for the same initial condition ($N = 30$) for *repulsive* interactions given by $g = 0.4, 0.54$, and $g = 1$, respectively. (**e,f**) Trajectories of phase singularities for an initial condition with $N = 5$ in the linear case and for *attractive* interactions given by $g = -0.75$, respectively. For $N = 5$ the merging time in the non-interacting case is $t_m = 3.4$ a.u. an decreases with $|g|$ (see Figure 2). Green (red) tubes represent positively (negatively) charged singularities.

The vortex lines shown in Figure 6b, for $g = 0.4$ illustrate how merging time is increased with $g$. The case plotted in Figure 6c, calculated for $g = 0.54$, shows a case where the singularities do not merge and seem to stay parallel to the time axis for the whole computational time (here only plotted up to $t = 4$ a.u. to facilitate location of merging time). For larger $g$, the singularities seem to travel outwards (e.g., as in Figure 6d, for $g = 1$). However, when calculating for longer times we observe that they bend inwards again. Nevertheless we cannot study longer dynamics, because of the presence of the computational boundary, so we cannot be conclusive about the long-time behavior of the singularities.

In Figure 6e,f we plot the position of the two singularities as a function of time for the non-interacting case and for attractive interactions, when $g = -0.75$, with $N = 5$. In the attractive case, for $N = 30$ instability occurs for very small interactions before merging time, so we calculated for various cases with $N < 20$ (see also Figure 7). For $N = 5$ the merging time in the non-interacting case is $t_m = 3.4$ a.u. (see Figure 5). As seen in Figure 6f for attractive interactions and $N = 5$ the merging time is decreased. Nevertheless, this is not always the case, as we discuss below.

For better visualization, we plot in Figure 7a the $x$-projection of the trajectories of the singularities for the same cases as in Figure 6a–d, that is, for $N = 30$ and for $g = 0, 0.4, 0.54$ and 1 (blue, magenta, red, green curves respectively). Here, we also see the behavior of the cases at $g = 0.54$ and $g = 1$, where the singularities seem to travel parallel or outwards, respectively. Nevertheless, as said before, we cannot conclude that they stay traveling parallel or outwards due to computational limitations. We plot in Figure 7b the merging time as a function of $g$ for $N = 5$, $N = 20$ and $N = 30$ for the repulsive case. We see that

for $N = 30$ the merging time increases only or $g > 0.38$ approximately while for $N = 5$ and $N = 20$ it increases for every $g > 0$.

In Figure 7c, we plot the $x$-axis coordinate of the singularities for the attractive case, with $N = 5$ and for $g = 0, -0.25$, and $-0.75$. As seen, the merging time decreases as $|g|$ is increased. Nevertheless, this behavior changes as $N$ is increased. In Figure 7d we plot the merging time as a function of $|g|$ for $N = 5$, $N = 10$ and $N = 20$ for the attractive case. We see that for $N = 5$ (blue curve) the merging time is indeed reduced as the strength of interactions is increased. For $N = 10$ (green curve) instead it decreases slightly. For $N = 20$ (orange curve) the merging time increases with the strength of interactions. It is not the goal of this paper to study this effect in depth. We conjecture this is an effect of a non-trivial interplay which involves the dynamical behavior of the density.

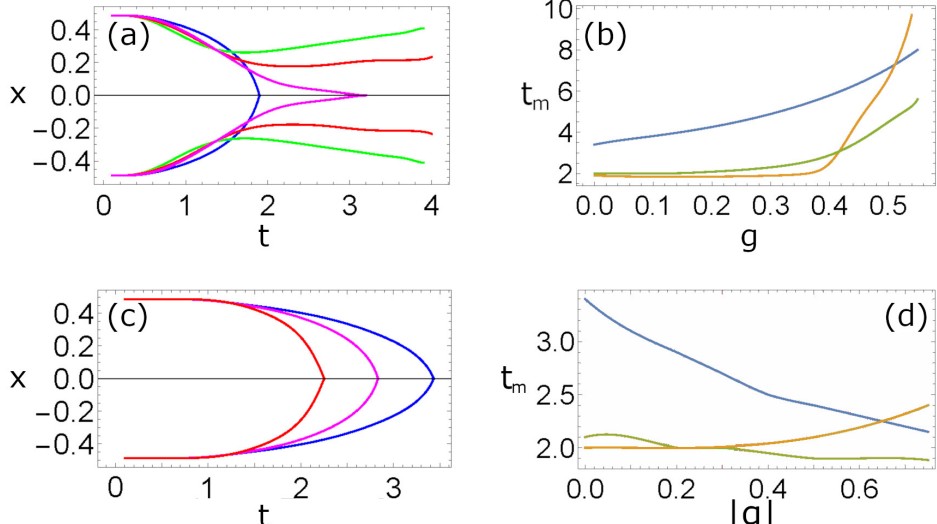

**Figure 7.** (**a**) Representation of the $x$ position of the trajectories with time, in the repulsive case, for the same four exemplary cases as in Figure 6 ($N = 30$, blue, magenta, red, green curves for $g = 0, 0.3, 0.54$ and $g = 1$, respectively). As shown, for $g = 0.54$ they stay parallel for long times. (**b**) The merging time as a function of $g$, for $N = 5$ (upper, blue curve), $N = 20$ (green middle curve), and $N = 30$ (lower, orange curve) atoms in the repulsive case. For $N = 5$ the merging time is increased for any value of $g$. On the contrary, for $N = 30$, only for $g$ larger than 0.38 the merging time starts to grow. (**c**) same than (**a**) for the attractive case, and three exemplary cases ($N = 5$, blue, magenta, red curves for $g = 0, -0.25$ and $g = -0.75$, respectively). (**d**) The merging time as a function of $|g|$, for $N = 5$ (blue curve), $N = 10$ (green curve), and $N = 20$ (orange curve) atoms in the attractive case. For $N = 5$ the merging time decreases with $|g|$. For $N = 10$ it decreases slightly and for $N = 20$ it increases. In all cases we show the results before instability occurs (we do not show the case with $N = 30$ as instability occurs already for $g = -0.3$).

### 3.2. Two Positive Singularities and One Negative

Now, let us consider the case with a negative singularity at the origin and two positives, one at $\mathbf{a_1} = (a_1, 0)$, the other at $\mathbf{a_2} = (a_2, 0)$, e.g.,

$$\phi(w, \bar{w}, 0) = (w - a_1)(w - a_2)\bar{w}\phi_{00}(w, \bar{w}). \tag{31}$$

For $g = 0$ this case was extensively detailed in [41]. We summarize here all the steps for illustrative purposes and because the parameters differ from the optics case. We expand initial condition (31), obtaining

$$\phi(w, \bar{w}, 0) = [w|w|^2 - (a_1 + a_2)|w|^2 + a_1 a_2 \bar{w}]\phi_{00}(w, \bar{w}). \tag{32}$$

From here (step 2) we obtain

$$\phi(w, \bar{w}, t) =$$
$$\phi_{11}(w, \bar{w}, t) - (a_1 + a_2)\phi_{01}(w, \bar{w}, t) + a_1 a_2 \phi_{-10}(w, \bar{w}, t). \tag{33}$$

Now we use Equation (22), to obtain the one wave function we do not have from previous example

$$\phi_{11}(w, \bar{w}, t) = w\left(\frac{i\sigma^2}{q(t)}\right)^2\left(\frac{t\sigma^2}{\pi q(t)}\right)(2 - \gamma(t)|w|^2)\exp\left[\frac{-i\pi|w|^2}{q(t)}\right]. \tag{34}$$

We obtain the solution from Equation (23) (step 4)

$$\phi(w, \bar{w}, t) =$$
$$\left(\frac{i\sigma^2}{q(t)}\right)^2 \exp\left[-\frac{i\pi|w|^2}{q(t)}\right]\left(w\left(\frac{t\sigma^2}{\pi q(t)}\right)(2 - \gamma(t)|w|^2)\right.$$
$$\left. - (a_1 + a_2)\left(\frac{t\sigma^2}{\pi q(t)}\right)(1 - \gamma(t)|w|^2) + a_1 a_2 \bar{w}\right). \tag{35}$$

Equation (35) represents the solution for all time in the linear case. We also solved numerically for interacting cases. In Figure 8 we present some illustrative cases, for $N = 150$. In panels (a) to (d) we present amplitude and phase for $g = 0.3$ at $t = 1$ a.u. and at $t = 4$ a.u. ((c) and (d)). As seen one positive singularity has merged with the central negative singularity, leaving in the origin a positive singularity. We plot in (e) to (h), amplitude and phase for $g = 0.6$ at the same times. Very interestingly, now we observe that at $t = 4$ a.u. merging has not yet occurred. Very remarkably, two pairs of positive/negative have appeared far from origin. We discuss this effect succinctly after obtaining the vortex lines.

Then, to look in some more depth the behavior of the phase singularities, let us study the trajectories in the linear case. From Equation (35) we find the equations for the trajectories of the singularities and the merging point. For this particular case the equation is

$$wt\sigma^2(2 - \gamma(t)|w|^2) - (a_1 + a_2)t\sigma^2(1 - \gamma(t)|w|^2)$$
$$+ a_1 a_2 \bar{w}\pi q(t) = 0. \tag{36}$$

To find the three zeros we need also to solve the complex conjugate of this equation. In the case that $a_2 = -a_1$ the evolution is such that at certain merging time $t_m$ one positive charged singularity merges with the central negatively charged one. The merging time is

$$t_m = \frac{a_1^2 2\pi\sigma^2}{\sqrt{16\sigma^4 - 4a_1^4\pi^2}}. \tag{37}$$

In Figure 9 we plot the position of the three singularities as a function of time (the vortex lines), for $a_1 = 1$, $a_2 = -1$, and $N = 150$ atoms (giving $\sigma^2 \approx 10.5$) for (a) the linear $g = 0$ case and two exemplary interacting cases, (b)–(d) with $g = 0.3$, $g = 0.45$ and $g = 0.6$, respectively. Again, we use the analytical method for the linear case and numerical split-step simulations both for the linear (to check it coincides with the analytical) and interacting cases. The analytically calculated merging time is now $t_m = 1.9$ a.u., and again it corresponds with the numerically calculated one. As in previous example we observe that for increasing interactions, the merging time gets larger and larger. We observe a second effect which is that the singularities emerge again after certain time (see panel (b) for $g = 0.3$. We have calculated for a collection of values between $g = 0.3$ and $g = 0.6$. We observed that this re-emergence of the singularities occurs closer and closer to the merging time and eventually the singularities do not merge again in very long calculations (we calculated for values pf $g$ up to 1). Also, we observed a third effect of the interactions, really

remarkable: at certain points, vortex/antivortex pairs appear. They occur in areas of low density, but outside of the central ring. From Figure 8c, one see that for $g = 0.3$ the low density area occurs away from origing. This is the area where the singularities appear in Figure 8g,h for a larger interactions, $g = 0.6$. This effect is then not at all trivial, but it falls out of the scope of this paper. It will be the subject of future research. Finally, we mention that in this example we have fixed the positions of the vortices in the initial condition in the $x$ axis. To consider arbitrary positions for the two positive singularities is possible, but it provides lengthly expressions that we do not reproduce here.

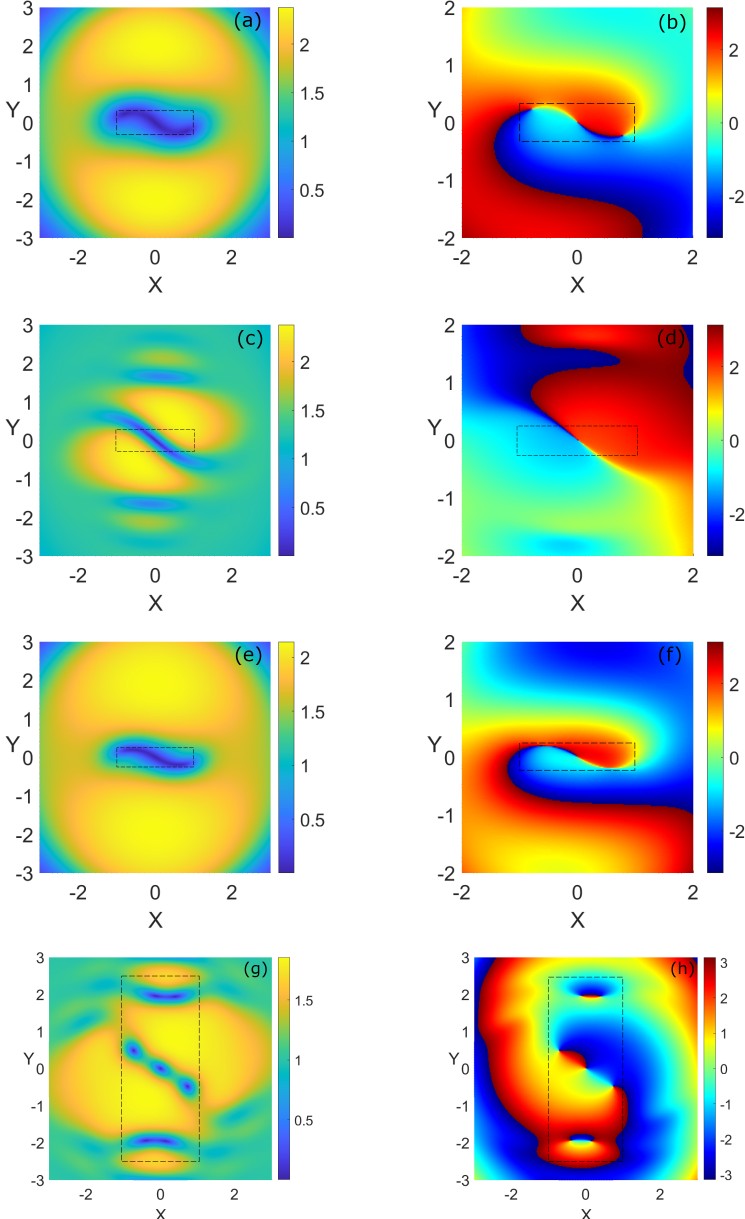

**Figure 8.** For $N = 150$, amplitude and phase for $g = 0.3$ after $t = 1$ a.u. (**a**,**b**) and after $t = 4$ a.u. (**c**,**d**). In (**d**) we see that merging of two of the singularities has occurred, leaving only one on-axis singularity. (**e**–**h**), same for $g = 0.6$. Now, at $t = 4$ a.u. merging has not yet occurred. Also two pairs of singularities have appeared far from origin. The dashed black squares mark the plotting box in Figure 9.

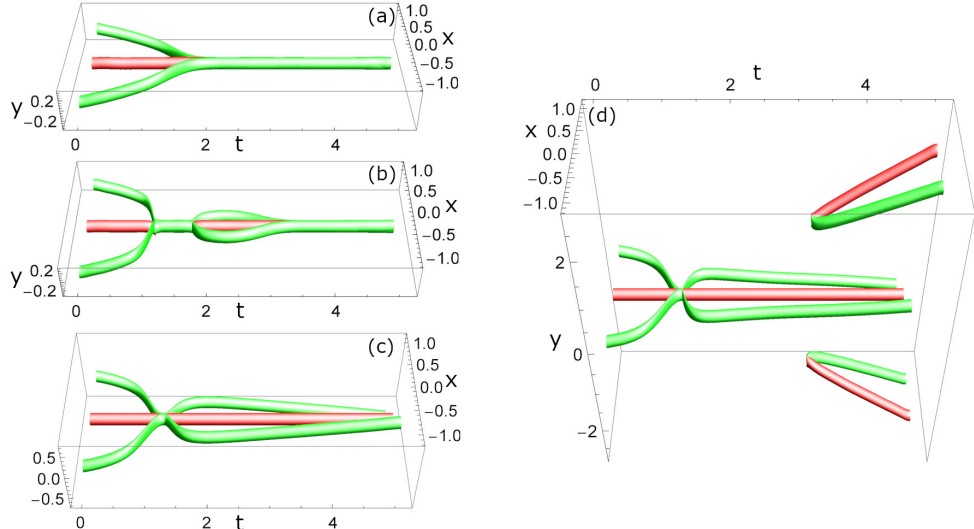

**Figure 9.** (**a**) Trajectories of phase singularities when the initial condition contains a phase singularity of charge $q = -1$ at the origin, and two singularities with $q = 1$ each one, at $\mathbf{a} = (\pm, 1, 0)$, when (**a**) $g = 0$, (**b**) $g = 0.3$, (**c**) $g = 0.45$ and (**d**) $g = 0.6$, when $N = 150$. For the non-interacting case, the merging point coincides with the analytically calculated. The interactions move the merging time (**b**). There is also a re-appearance of the singularities (see panels (**b**,**c**)). For large enough singularities, these singularities do not merge again for very long simulations (see panel (**d**)). There is also one more effect visible in panel (**d**): the generation of vortex-antivortex pairs. Green (red) tubes represent positively (negatively) charged singularities.

## 4. Some Examples in the Parabolically Trapped System

In this section we illustrate the dynamical evolution of initial conditions containing vortices when the external potential is a parabolic trap. The first example is a single singularity located outside the center of the potential trap, i.e.,

$$\phi(w, \bar{w}, 0) = (w - \mathbf{a})\phi_{00}(w, \bar{w}). \tag{38}$$

In the second example the initial condition contains two singularities, both away from the center of the potential trap, one positively charged and one negatively charged,

$$\phi(w, \bar{w}, 0) = (w - \mathbf{a})(\bar{w} - \mathbf{b})\phi_{00}(w, \bar{w}). \tag{39}$$

In the third example the initial condition contains three singularities, one negatively charged located in the minimum of the potential, and two away from its center, both positively charged,

$$\phi(w, \bar{w}, 0) = (w - \mathbf{a})(w - \mathbf{a}')\bar{w}\phi_{00}(w, \bar{w}). \tag{40}$$

To solve analytically in the linear $g = 0$ case, we use the method described in Section 2, for the trapped case. The main difference with the homogeneous case is that one propagates $\phi_{00}$ using propagator (19) to obtain Equation (20). In Figure 10 we present the dynamical evolution of the singularities for the three examples (we emphasize that in all cases we checked that energy is conserved). In Figure 10a we present the case of a single phase singularity, when $\mathbf{a} = (1, 0)$ and $N = 12$ ($\sigma^2 = 10$). The trajectory shows the influence of the trap, as it spins around approaching more the center of the trap. For the second example, shown in Figure 10b, the trajectories of the two phase singularities with opposite charge bend around and eventually merge and annihilate each other. From that time on, no singularity persists in the field. Here, $\mathbf{a} = (1, 0)$, $\mathbf{b} = (-1, 0)$ and $N = 30$ ($\sigma^2 = 10$). In the third example (Figure 10c), the positively charged singularities perform a spiral movement before reaching the center of the trap, where one of them annihilate with the

central negatively-charged one. From that time on, there is only one phase singularity in the field which stays in the potential minimum of the trap. Here, $\mathbf{a} = (1.5, 0)$, $\mathbf{a}' = (-1.5, 0)$ and $N = 150$ ($\sigma^2 = 10$). These three examples show the rich variety of dynamics one can explore with this method in a parabolic trap. In the panels above these figures we present these evolutions as three dimensional plots, for a complementary visualization.

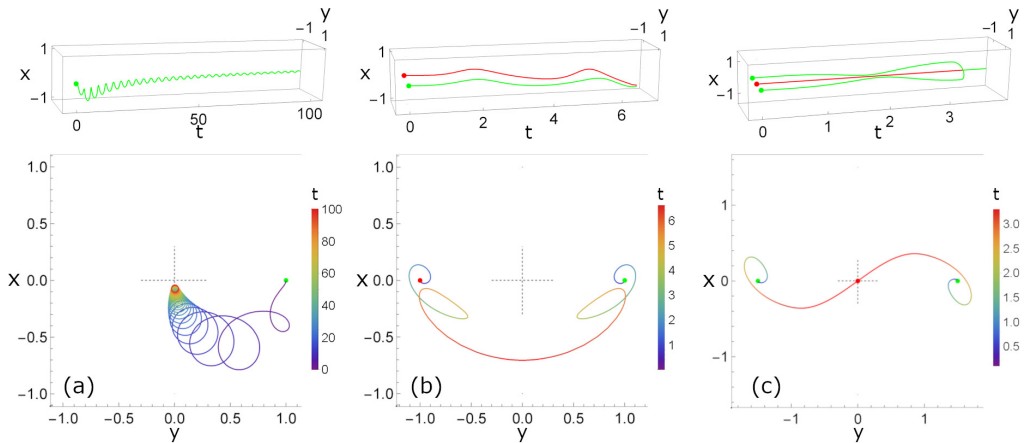

**Figure 10.** (**a**) Trajectory of a single phase singularity in a parabolic trap, initially off-axis, i.e., located at $\mathbf{a} = (1, 0)$, with $N = 12$ atoms. (**b**) Trajectories of two phase singularities in a parabolic trap, one positively charged initially at $\mathbf{a} = (1, 0)$ and one negatively charged initially at $\mathbf{b} = (-1, 0)$, with $N = 30$ atoms. In this case, the singularities merge at a $t_{\mathrm{m}}$ which we obtain numerically. (**c**) Trajectories of a negatively charged phase singularity initially at the center of the trap, and two positively charged phase singularities initially at $\mathbf{a} = (1.5, 0)$ and $\mathbf{a}' = (-1.5, 0)$, respectively, in a potential parabolic trap, with $N = 150$ atoms. The two positively charged singularities tend to the center of the trap, where one of them annihilates the central negative phase singularity leaving only one positively charged singularity which stays there for the rest of the evolution. In all cases, $g = 0$. In all panels, time is represented with a color gradient from blue to red. Green (red) circles represent positively (negatively) charged singularities. Also, we plot on top of each panel the evolution in time, for better visualization.

To consider interactions shows similar effects as in the homogeneous case, that is, to contribute to avoiding annihilation of singularities or to create pairs, for instance. To illustrate this, we show in Figure 11 the evolution with $g = 0$ and 0.3 when $\mathbf{a} = (0.5, 0)$ and $\mathbf{b} = (-0.5, 0)$. This is similar to the second example (shown in Figure 10b), but we put the singularities initially closer to the minimum of the potential to make the merging time shorter. As shown in Figure 11a, the singularities tend to center of the trap as in Figure 10b, and annihilate each other at a merging time of $t_{\mathrm{m}} = 1.5$ a.u., leaving a field without singularities from that time. In the interacting case, shown in Figure 11b, they do not annihilate at this merging time. Instead, the singularities repel, and perform complicated trajectories around the center of the trap from that time on. We also observe creation of pairs which eventually annihilate. The results for larger $g$ (not shown) are qualitatively similar to the homogeneous case, that is, merging in the center of the trap is avoided and pairs are created. Here, the trajectories become very intricate and for that reason we do not include them.

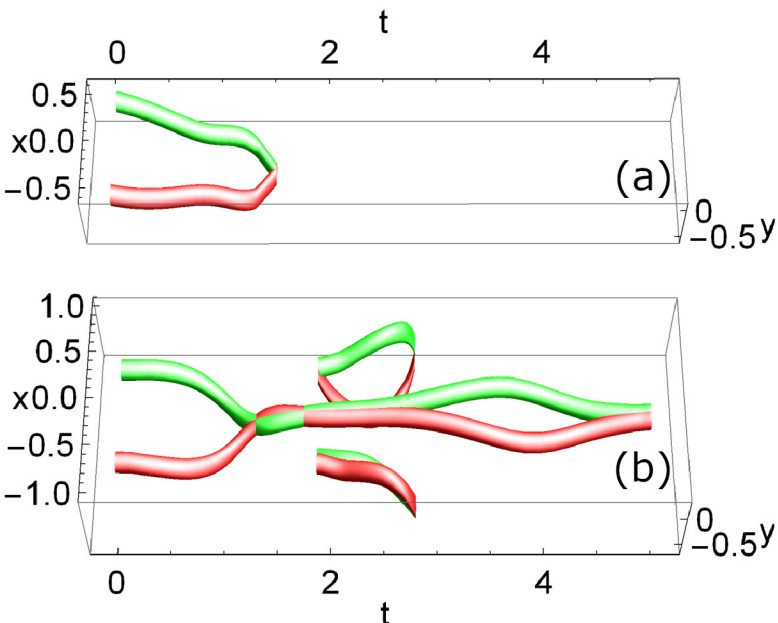

**Figure 11.** Trajectory of two phase singularities in a parabolic trap, one positively charged initially at **a** $= (0.5, 0)$ and one negatively charged initially at **b** $= (-0.5, 0)$, for $N = 30$ atoms, and (**a**) $g = 0$ and (**b**) $g = 0.3$. In the linear case the singularities merge at a merging time of $t = 1.2$ a.u. In the interacting case, the singularities tend to the origin but do not merge. Instead, after colliding at the center of the trap they perform an intricate dynamics. Green (red) tubes represent positively (negatively) charged singularities.

## 5. Conclusions and Outlook

We presented a method which allows one to study different aspects of the dynamics of a quasi-two dimensional Bose-Einstein condensate with a number of vortices located at arbitrary points. The method has an analytical part, which is only valid when one considers a vanishing coupling constant. We considered two possible realizations: a homogeneous system and a Bose-Einstein condensate in a parabolic trap. For the homogeneous system, the method is similar to that introduced in the context of photonics [41]. Here, we adapted it to the case of atoms. In this paper, we extended the method to the trapped case, which is more conventional in the context of Bose-Einstein condensates. This requires us to determine a generating function, which is very different than the one in the homogeneous case.

Once this first part is summarized in a recipe with four steps, we showed that the analytical solution can be used to determine the trajectories of the singularities (vortex lines), via a set of algebraic equations. These equations can be used to determine some figures of merit. Then, for the non-linear case, it is necessary to find the numerical solution of the Gross-Pitaevskii equation via a split-step simulation. The role of the analytical solution found in the first part is to determine a non-interacting scenario to compare with. The numerical simulation of the non-linear case allows one to observe how interactions modify the dynamics obtained in the non-interacting case. For small interactions and depending on the number of atoms, the dynamics does not change appreciably in some cases.

We illustrated the method with several examples. In the homogeneous system, we discussed the cases of two singularities with opposite charge and three singularities, one, located at the origin, with opposite charge than the other two. We found the equations for the trajectories of the singularities in the two cases. With this we found the merging time at which two singularities of opposite charge will collide and annihilate. We then turned on interactions, both attractive and repulsive, and found, numerically, how the trajectories change and how the merging time changes as a function of the number of particles and the coupling constant. For two singularities, we exemplified the kind of study one can perform numerically with this method. For repulsive interactions, we observed that the number of

particles reduces the merging time for the non-interacting case. The merging time increases with the coupling constant, but for larger $N$ the merging time is less sensitive to small values of the coupling constant. For $N = 30$ atoms we observed that the merging time does not change until $g = 0.38$. On the contrary, for larger $g$ the behavior seems similar irrespectively of the number of atoms. For attractive interactions we see that the behavior is very affected by the presence of the instability. We show that there is a complex interplay with the density, which for small $N$ makes the merging time decrease with the strength of the interactions and for larger $N$ makes the merging time increase with the strength of the interactions. For three singularities, we observed qualitatively the effect of interactions (only repulsive) in merging time and an additional effect of the interactions, which is the spontaneous creation of vortex/antivortex pairs at certain times.

For the inhomogeneous, trapped system, we included examples for one, two oppositely charged singularities, and three singularities, the one in the origin with opposite charge than the other two. We showed, within the non-interacting scenario, that the trajectories are influenced by the trap, making the trajectories spiral-like in the trap. We illustrated the introduction of interactions in one case, only to show that similar effects as in the homogeneous case are observed, that is, modification of merging time and creation of vortex/antivortex pairs.

The results presented in this paper are only intended to illustrate the utility of the method. We envisage several future research directions. For example, to study systematically the dependence of different quantities of interest, not only merging time, with number of atoms and interactions, for different cases, such as those included here, both in the homogeneous and trapped cases. We also consider it interesting to study the phenomena of creation of vortex/antivortex pairs and its dependence with number of atoms, coupling constant, and trap frequency. Another research problem which can be approached with this method is to study initial conditions with many singularities at random positions; we note that this last case will require us to determine its locations numerically (which as commented requires a dedicated numerical method as [84,85]). In addition, it will require us to solve a large set of equations even in the non-interacting case. The perturbative study and comparison with the analytical solution may give some hints as well. Last but not least, we see that one can pursue adapting the method to initial conditions which are more conventional in the context of BECs. That is, with a Thomas-Fermi profile and vortices separated by their vortex cores or Jones-Roberts solitons sustaining vortices as in [78,79]. The comparison with the effective models established in the literature may give interesting results. For the initial conditions studied here, it is still of interest to study the interplay of the singularities with the density, probably with similar models as those used for initial conditions with a Thomas-Fermi profile. All these possible directions show that this method can be of utility in the study of vortices in Bose-Einstein condensates.

**Author Contributions:** Conceptualization, A.F. and M.Á.G.-M. methodology, A.F. and M.Á.G.-M.; software, S.D.M.-G.; validation, S.D.M.-G. and M.Á.G.-M.; formal analysis, all authors; investigation, all authors; resources, A.F., P.F.D.C. and J.A.C.; writing—original draft preparation, M.Á.G.-M. and S.D.M.-G.; writing—review and editing, all authors; visualization, S.D.M.-G.; supervision, A.F., P.F.D.C., J.A.C. and M.Á.G.-M.; project administration, A.F., P.F.D.C. and J.A.C.; funding acquisition, A.F., P.F.D.C. and J.A.C. All authors have read and agreed to the published version of the manuscript.

**Funding:** M.Á.G.-M. acknowledges funding from the Spanish Ministry of Education and Professional Training (MEFP) through the Beatriz Galindo program 2018 (BEAGAL18/00203), Spanish Ministry MINECO (FIDEUA PID2019-106901GBI00/10.13039/501100011033), QuantERA II Cofund 2021 PCI2022-133004, Project of MCIN with funding from European Union NextGenerationEU (PRTR-C17.I1) and by Generalitat Valenciana, with Ref. 20220883 (PerovsQuTe). P.F.D.C. acknowledges the grant PID2021-128676OB-I00 funded by FEDER/MCIN. J.A.C. acknowledges funding from the Spanish Ministry of Science and Innovation (MICINN), grant PID2021-124618NB-C21. A.F. thanks the support of MCIN of Spain through the Project No. PID2020-120484RB-I00 and Generalitat Valenciana, Spain (Grant No. PROMETEO/2021/082).

**Data Availability Statement:** Not applicable.

**Acknowledgments:** The authors would like to acknowledge the useful and insightful comments from all referees. M.Á.G.-M. acknowledges funding from the Spanish Ministry of Education and Professional Training (MEFP) through the Beatriz Galindo program 2018 (BEAGAL18/00203), Spanish Ministry MINECO (FIDEUA PID2019-106901GBI00/10.13039/501100011033), QuantERA II Cofund 2021 PCI2022-133004, Project of MCIN with funding from European Union NextGenerationEU (PRTR-C17.I1) and by Generalitat Valenciana, with Ref. 20220883 (PerovsQuTe). P.F.D.C. acknowledges the grant PID2021-128676OB-I00 funded by FEDER/MCIN. J.A.C. acknowledges funding from the Spanish Ministry of Science and Innovation (MICINN), grant PID2021-124618NB-C21. A.F. thanks the support of MCIN of Spain through the Project No. PID2020-120484RB-I00 and Generalitat Valenciana, Spain (Grant No. PROMETEO/2021/082).

**Conflicts of Interest:** The authors declare no conflict of interest.

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
