# Peer review of "A Method for the Dynamics of Vortices in a Bose-Einstein Condensate: Analytical Equations of the Trajectories of Phase Singularities"

_condensedmatter, doi:10.3390/condmat8010012_

Round 1

Reviewer 1 Report

This manuscript demonstrates the use of a method for predicting the dynamics of vortices in systems described by Schrodinger-like equations of motion, with a particular focus on atomic Bose-Einstein condensates. The method, summarised in Sec. II, was previously developed in Refs. [55,56], and the current work presents examples of how the method can be applied to the nonlinear Bose-Einstein condensate system. The authors present a number of vortex trajectories that have been calculated using the method using a few simple initial configurations. They vary the interaction strength and add a harmonic trapping potential to see how the dynamics are affected.

I had a number of questions about the calculated vortex trajectories that were not discussed by the authors. If these can be addressed satisfactorily, I can recommend publication in Condensed Matter.

Main comments:
1. Have you confirmed that the numerical simulations conserve energy and normalisation to reasonable accuracy? And have you compared the trajectories calculated from this method to trajectories obtained from more conventional methods of numerically solving the Gross-Pitaevskii equation? I think checks like these are necessary to demonstrate the validity of the method.
2. In Fig 1c/d, why do the vortices separate for larger interaction strength g? I would expect them to travel in a straight line (perpendicular to their separation vector) in the limit g->infinity.
3. Regarding the number of atoms N:
   (i) In Fig 1f, why does the merging time vary with the number of atoms in the noninteracting case? The equation of motion (1) reduces to the linear Schrodinger equation, for which the normalisation of the wavefunction does not affect the dynamics. What is the physical interpretation of the wavefunction norm in the case of an optical system?
   (ii) Could the authors explain how N determines the width sigma?
4. In Fig 2b/c, how is pair creation possible here? I would not expect it to be energetically favourable, given the kinetic energy cost of creating vortices. Could the authors explain this?
5. In Fig 3a, why does the vortex end up circling a point away from the trap centre? How does this trajectory conserve energy? I would expect that the energy is lower when the vortex is further from the trap centre, but perhaps my intuition is incorrect in this situation.
6. In all figures, what are the vortex core sizes? How do they compare to other relevant length scales like the trap radius, or (where there are multiple vortices) the inter-vortex spacing? Do the cores overlap?
7. Regarding Fig 1:
   (i) Are the results in Fig 1a-f all numerical?
   (ii) Should the x-axis of Fig 1f be labelled "N", rather than "g"?
   (iii) What is the physical significance of the sudden increase in the merging time for N=20 for g~0.38, as pointed out in the text?
   (iv) It would be useful to point out in the caption that (f) is plotted for the case g=0.
8. Are the results in Fig 2a-c all numerical?
9. Regarding Fig 3:
   (i) Are all the results numerical?
   (ii) Are all trajectories shown for g=0?
   (iii) Could you somehow show the edge of the trap is these figures? e.g. a dotted line where the density falls to 0.05 of the maximum value?
   (iv) It would be useful to plot these trajectories on equally spaced axes (so that the trap is circular).

Minor comments:
- Eq (1): the subscript "t" for the Laplacian is ambiguous, since "t" is also used for time. I suggest the authors change the notation here.
- Eq (14): the symbol "t" is used again, but it seems to be an amplitude, rather than time. I suggest the authors also change this notation to avoid confusion.
- Following Eq (18), the authors refer to "\psi_{0,0}" - should this be "\phi_{00}"?
- In Eqs 19-20, both "*" and "." symbols appear to be used to denote multiplication. I suggest using one symbol for consistency.
- Immediately following Eq (24): "the results obtained are long and we omit them here". If I understand correctly, here you are considering one vortex and one antivortex in a uniform 2D system. In that case, your result should be quite general, as I would expect that only the distance |a-b| determines the dynamics. Is this not the case here? Perhaps the vortex core size plays a role?
- In the paragraph below Eq (30), the merging time is quoted in "a.u.". Does this correspond to units of 1/omega?
- A comment has been left in the caption of Fig 4, which should be removed.
- I do not understand the first sentence of the last paragraph of Sec IV ("To consider interactions..."). I suggest rephrasing this sentence to clarfiy what is meant.

I also found a few minor typos throughout the macnuscript:
- The authors use the word "count" twice in the first paragraph, but I am not sure it is being used correctly.
- The word "y" is used in a few places throughout the paper - presumably these should be replaced with "and".
- Immediately following Eq (6): "plane" should be "plain".
- Immediately before Eq (14): "modees" should be "modes".
- Immediately following Eq (15): "use this commutators" should be "use these commutators"
- Fourth-last sentence in Sec IIIA: "only or g>0.38" should be "only for g>0.38".
- Last sentence on page 6: "lengthly expressions we so" should be "lengthy expressions we do".

Author Response

This manuscript demonstrates the use of a method for predicting the dynamics of vortices in systems described by Schrodinger-like equations of motion, with a particular focus on atomic Bose-Einstein condensates. The method, summarised in Sec. II, was previously developed in Refs. [55,56], and the current work presents examples of how the method can be applied to the nonlinear Bose-Einstein condensate system. The authors present a number of vortex trajectories that have been calculated using the method using a few simple initial configurations. They vary the interaction strength and add a harmonic trapping potential to see how the dynamics are affected.

AUTHORS COMMENT: we thank the referee for his/her constructive report.

I had a number of questions about the calculated vortex trajectories that were not discussed by the authors. If these can be addressed satisfactorily, I can recommend publication in Condensed Matter.

Main comments:

COMMENT 1. Have you confirmed that the numerical simulations conserve energy and normalisation to reasonable accuracy? And have you compared the trajectories calculated from this method to trajectories obtained from more conventional methods of numerically solving the Gross-Pitaevskii equation? I think checks like these are necessary to demonstrate the validity of the method.

AUTHORS REPLY 1: We have calculated the energy and it is conserved for all the cases. We don't include any figure because it will be a straight constant line, but we have explained it in the new ms. We have not compared our results with other cases, but the split-step method is usually used to solve the non-linear Groos-Pitaevskii equation (e.g., see the appendix B of Nonlinear Fiber Optics https://www.elsevier.com/books/nonlinear-fiber-optics/agrawal/978-0-12-397023-7).

COMMENT 2. In Fig 1c/d, why do the vortices separate for larger interaction strength g? I would expect them to travel in a straight line (perpendicular to their separation vector) in the limit g->infinity.

AUTHORS REPLY 2: We can not guaranteed that the distance between the singularities remains constant o increase to time large enough, e.g., in figure 2c when g = 0.54 we have seen that the singularities travel parallels, but finally they end up coming together. We have conjectured that it is result of the limit of our numerical simulations (see discussion of Fig. 3 and 4 in ms).

COMMENT 3. Regarding the number of atoms N:

   (i) In Fig 1f, why does the merging time vary with the number of atoms in the noninteracting case? The equation of motion (1) reduces to the linear Schrodinger equation, for which the normalisation of the wavefunction does not affect the dynamics. What is the physical interpretation of the wavefunction norm in the case of an optical system?

   (ii) Could the authors explain how N determines the width sigma?

AUTHORS REPLY 3: The merging point changes because the ring of the initial condition is different depending on N (see figure 1 and the appendix of the new ms), then if we have more density the interaction between singularities will be stronger and the merging point occurs sooner – what changes is the shape of the initial condition, as we write it in the form of Eq. (4) (see new Figs. 1 and 8). The normalization of the wave function norm in the optical systems is meant as the intensity of optical beam. The expression of sigma depending on the N is different for the different initial conditions. We have calculated an approximation numerically and we have explained it in the new ms.

COMMENT 4. In Fig 2b/c, how is pair creation possible here? I would not expect it to be energetically favourable, given the kinetic energy cost of creating vortices. Could the authors explain this?

AUTHORS REPLY 4: We have some conjectures about this behavior. This is an interesting effect that occurs even in the linear case for photonic systems when changes on discrete symmetry are involved. We have not a high level explanation but we have checked that the energy of the system is conserved probably because this emission is inside the vortex ring density. We will study this topic in future research.

COMMENT 5. In Fig 3a, why does the vortex end up circling a point away from the trap centre? How does this trajectory conserve energy? I would expect that the energy is lower when the vortex is further from the trap centre, but perhaps my intuition is incorrect in this situation.

AUTHORS REPLY 5:We can calculate the temporal limit of the singularity analytically and see that it approaches asymptotic to the (0,0) – we also did as long-time calculations and we see it approaches the origin. See also new appendix B. The conservation of the energy does not depend only on the trajectory of the singularity, it involves the intensity of the ring too, that is why it remains constant when we calculate it.

COMMENT 6. In all figures, what are the vortex core sizes? How do they compare to other relevant length scales like the trap radius, or (where there are multiple vortices) the inter-vortex spacing? Do the cores overlap?

AUTHORS REPLY 6: Please, see the authors reply 1 of the referee 1.

COMMENT 7. Regarding Fig 1:

   (i) Are the results in Fig 1a-f all numerical?

   (ii) Should the x-axis of Fig 1f be labelled "N", rather than "g"?

   (iii) What is the physical significance of the sudden increase in the merging time for N=20 for g~0.38, as pointed out in the text?

   (iv) It would be useful to point out in the caption that (f) is plotted for the case g=0.

AUTHORS REPLY 7: The results of the figure 1 a-c (new figure 2 a-d) and the figure 2 d-e (new figure 3 a-b) are all numerical, but figure 1f (new figure 3 c) was calculated analytically. We always compare with the corresponsding simulation for the analytical cases.

The x-axis of figure 1 f was wrong, it is “N” rather than “g”. We thank the referee for point this out.

About the significance of the figure 1 e see the authors reply 3a of the referee 2.

Now figures are splitted in different ones, to make them more explanatory.

COMMENT 8. Are the results in Fig 2a-c all numerical?

AUTHORS REPLY 8: Yes, the results of the 3 singularities behavior have been calculated with the split-step method (see appendix)

COMMENT 9. Regarding Fig 3:

   (i) Are all the results numerical?

   (ii) Are all trajectories shown for g=0?

   (iii) Could you somehow show the edge of the trap is these figures? e.g. a dotted line where the density falls to 0.05 of the maximum value?

   (iv) It would be useful to plot these trajectories on equally spaced axes (so that the trap is circular).

AUTHORS REPLY 9: The trajectories of the figure 3 (new figure 5) were calculated with the analytical method explained in section 2 after the equation (22). These results show the behavior of the singularities in the linear case with a parabolic trap, so the value of g is 0 in all three cases. We have added the changes commented for the referee to make the figures more informative.

Minor comments:

- Eq (1): the subscript "t" for the Laplacian is ambiguous, since "t" is also used for time. I suggest the authors change the notation here.

- Eq (14): the symbol "t" is used again, but it seems to be an amplitude, rather than time. I suggest the authors also change this notation to avoid confusion.

- Following Eq (18), the authors refer to "\psi_{0,0}" - should this be "\phi_{00}"?

- In Eqs 19-20, both "*" and "." symbols appear to be used to denote multiplication. I suggest using one symbol for consistency.

- Immediately following Eq (24): "the results obtained are long and we omit them here". If I understand correctly, here you are considering one vortex and one antivortex in a uniform 2D system. In that case, your result should be quite general, as I would expect that only the distance |a-b| determines the dynamics. Is this not the case here? Perhaps the vortex core size plays a role?

- In the paragraph below Eq (30), the merging time is quoted in "a.u.". Does this correspond to units of 1/omega?

- A comment has been left in the caption of Fig 4, which should be removed.

- I do not understand the first sentence of the last paragraph of Sec IV ("To consider interactions..."). I suggest rephrasing this sentence to clarfiy what is meant.

I also found a few minor typos throughout the macnuscript:

- The authors use the word "count" twice in the first paragraph, but I am not sure it is being used correctly.

- The word "y" is used in a few places throughout the paper - presumably these should be replaced with "and".

- Immediately following Eq (6): "plane" should be "plain".

- Immediately before Eq (14): "modees" should be "modes".

- Immediately following Eq (15): "use this commutators" should be "use these commutators"

- Fourth-last sentence in Sec IIIA: "only or g>0.38" should be "only for g>0.38".

- Last sentence on page 6: "lengthly expressions we so" should be "lengthy expressions we do".

ARTHORS REPLY: we thank the referee for all these comments, which we corrected in new version.

Reviewer 2 Report

See attached file

Author Response

The authors adapt a method utilised in homogeneous photonic systems to study the dynamics of vortices in homogenous and trapped non-interacting BECs. The authors state that, “The results presented in this paper are only intended to illustrate the utility of the method”. Since the method applied to BECs and photonics is mathematically identical in the homogeneous case and only slightly modified in the trapped case, I do not think this is significant enough to warrant publication. However, a more thorough analysis and interpretation of the trapped dynamics and the effect of interactions would make the paper sufficiently interesting.

AUTHORS REPLY: We sincerely thank the referee for his/her careful reading and assessment of the manuscript. We have followed his/her recommendations and included new calculations, figures and changed the text when necessary. Particularly we have clarified the kind of initial condition we are considering and its difference with that usually considered in the literature. We have included calculations not only with repulsive, but also with attractive interactions. We have included a new appendix with the plots of density and phase of the evolved wave function to illustrate the reach of our method and also, somehow, its limitations so far. Please find below a detailed list with the replies to his/her comments, and see also replies to other referees.

REFEREE COMMENT 1: Can the authors discuss the relevance of their results to realistic interaction strengths and BEC experiments, in both the homogeneous and trapped cases? Presuming the results in Fig. (1) (d) are correct, can the method shed light on the collision of two vortices with overlapping cores? Can the authors extend the analytic method to shed light on the cross-over from non-interacting to weak (perturbative) interactions?

AUTHORS REPLY 1: We have done an effort in the new manuscript to explain clearly the adimensionalization we use, and with that, to clarify the range of validity of our results (see comment after Eq. (24)). We have also clarified the role of the number of atoms, considering that gN is a parameter that determines the regime of validity of the GPE, which was not clear in previous version (see reply to same referee comment below). We have calculated with a wider range of interactions (see Figures 3 and 4). Also for different signs of the interactions (not only repulsive but also attractive). For the discussion on vortices with overlapping cores, please see reply to next comment. The question on perturbations, assuming the interactions are small, is interesting, but we believe it will not add anything new to the conclusions and aims of this paper, as for small interactions there is small changes in merging points, and the differences from the non-interacting case occur far from perturbative regime. We leave it for future research as mentioned in conclusions. The results in the linear case are correct. The numerical results on the nonlinear case have been checked thoroughly (see below that energy is conserved, they reproduce the linear case, and the code has been checked in many other systems). In the new version, we discuss the numerical method in more detail (which is a conventional spectral method) so the community can more easily reproduce calculations (see pg. 8, first column). Motivated by other referee's comments, we have studied the long time behavior of the singularities, seeing if they would really depart from each other or eventually they merge at a larger time (see e.g. new discussion of Figs. 3 and 4).

REFEREE COMMENT 2: Sec III. Can the authors make a more thorough comparison with the well-known result that two oppositely charged vortices with separation greater than the healing length move in parallel lines? Presumably this emerges in the case of stronger interactions, which decreases the healing length

AUTHORS REPLY 2: We thank the referee for his/her comment. Thanks to this and other referee's comments we found that we have not been clear enough when explaining our aims in this manuscript. The initial condition in equation (4) is not that of a set of dark vortices. It is instead what would be a bright vortex soliton with two or more singularities embedded close to the vortex core. To make this clear we have included them explicitly in the new Figure 1 and in appendix, Fig. 8. This is an initial condition which is more conventional in self-focusing (attractive) non-linearities (interactions), but which can be created in an experiment and propagated in self-defocusing non-linearities. We discuss now this aspect in section II and in appendix A. We also included new calculations with attractive interactions. We finally discuss in outlook that it would be interesting for future work to extend the work to the initial conditions where vortices are embedded in a BEC, that is in a constant or Thomas-Fermi parabolic profile. In the light of this, we discuss in introduction and conclusions, how this mathematical tool contextualizes in present literature and how it can be extended to complement it or enrich present results. To be explicit, since the singularities are embedded within the same vortex core it is not possible to compare with the result that two oppositely charged vortices with separation greater than the healing length move in parallel lines, mentioned by referee. We comment this now also in the paper and cite accordingly.

REFEREE COMMENT 3: The introduction is quite broad and does not clearly give focus to the purpose of the manuscript.

AUTHORS REPLY 3: We have extended the introduction to include a more focused view of the literature on dynamics of singularities in BECS, which is an extensive literature. Now we include a new paragraph in introduction with a discussion of the topic (see last paragraph of introduction). Among theoretical and experimental works, we included more than 40 new references. Undoubtedly, we may miss some relevant references.

REFEREE COMMENT 4: The three dimensional figures are confusing. Are they necessary in Fig. 1?

AUTHORS REPLY 4: We have changed the point of view of the figures to make them more clear, also motivated by other referees comments. Taking into account comment 2, one could see that they are very interesting because all physics occur in the center of the simulation, close to the vortex core where the singularities are located.

Important technical modifications:

  1. Fig 3(a). I was surprised to see that a single vortex approaches the centre of the trap. Does this conserve energy? (In an interacting BEC, a single vortex circulates the centre of the trap, see for example, “Motion of vortices in inhomogeneous Bose-Einstein condensates” [Phys. Rev. A 97, 023617, 2018].)

AUTHORS REPLY: We have calculated the energy along all evolutions to make sure that it is conserved, as it is. We do not include the figures because they are only straight constant lines. One has to take into account that not only the singularities do evolve but also the density. To illustrate this we included an appendix where we show the evolution of the density after some time has passed. Also, we show here the main difference between repulsive and attractive interactions, which is in the behavior of the density. For the attractive case after some time instability occurs in many of the instances. We have included the reference suggested by the referee, which we find very interesting, to show the different system we are considering.

  1. In various points in the manuscript the authors consider the affect of changing g and changing N.

N from both sides of Eq. (3), and noting that g and N now only appear as a product in the interaction term,

These are essentially the same (as can be seen by normalizing φ to one, cancelling a factor of

gN|φ|2).

AUTHORS REPLY: We have included a clarification on the document about the role of gN (see text after Eq. (5)). The referee correctly points out that gN is the appropriate parameter when considering condensation. Here nevertheless in view of initial condition (4) we prefer to keep N as a normalization constant. It is of particular relevance for the self-attractive case, where self-focusing will depend on N. But also, we keep it to highlight its role in the merging time in the non-interacting case, so we prefer to have it explicitly.

  1. The choice of units in figures is not clear to me, and some axes are missing labels.

AUTHORS REPLY: We have corrected the missing labels, and we thank the referee for pointing this out. Also, we have included a more detailed discussion of the adminesionalization we use.

  1. Above Eq. (3) the authors state they ”we omit the hats in the operators”. Is this just a notation change, or are the authors implying a change from quantum to classical in these observables? In the limit of zero interactions, position and momentum commutation relations are still implied by the GPE formalism.

AUTHORS REPLY: It is just notation. We explicit this in the ms.

  1. The description of the numerical method to solve Eq. (1) does not provide sufficient detail for the results to be reproduced.

AUTHORS REPLY: We have referred now to a non-linear optics book, a topic where this method is used extensively. We have also explained succinctly the method and the parameters used in the new version of ms (see second column, page 6).

Reviewer 3 Report

Please find the referee report in the attached PDF file.

Author Response

In this paper, the authors present methods to study the vortex dynamics in quasi 2D Bose- Einstein condensates. An analytical solution was proposed to study the non-interacting scenario, for both the homogeneous and parabolic trapped cases, while a numerical split-step simulation was applied to the interacting scenario. The proposed methods were then used to study the time evolution in different vortex/singularity settings, showing some interesting merging and annihilation behaviors.

Overall, the presented results are novel and of interest for the researchers working in the field of Bose-Einstein condensates and vortex dynamics. Especially, the analytical solution for the non- interacting parabolic trapped case, was proposed for the first time in this field. I believe this manuscript meets the publication requirements of Condensed Matter, and I would recommend publication after minor revision.

AUTHORS REPLY: we thank the referee for his/her comments, which have been of great utility to improve the ms.

COMMENT 1. The trajectory figures are not well presented:

a. The 3D trajectories shown in Fig. 1 (a - c), 2(c) & 4 (a & b) are misleading, especially for

Fig. 1 (a - c) and Fig. 4 (a), it seems the center of vortices is moving along the x direction. Can the authors try to change the angle of the 3D plot to better display these trajectories? Another issue is that the trajectories in Fig. 1 (a) are thicker (see the size the red/green tubes) than those in other panels. Does this have any special meanings, if not, can the authors change them back to the same thickness?

b. All the figures are not well labeled. Different panels in each figure present different external field / interaction settings. Can the authors clearly label each panel, i.e., add subtitles for each panel, which will make the figures much more self-explanatory? Besides, the subplots in Fig. 3 do not have any axis labels (x, y and t). I’d recommend the authors to add those axis labels.

AUTHORS REPLY 1: We have changed the point of view and the thickness of the 3D figures to make them more clear, and we have plotted them in the same box to make easier the comparison of the different cases. We have separated figure 1 in new figures 2, 3 and 4, and changed the subtitles to make them more explanatory. We have added the missing labels too. The referee is right that the thickness does not have any meaning, so now all figures have same thickness.

COMMENT 2. Lacks numerical simulation details for the interacting scenario:

a. The authors provided a comprehensive description of the analytical methods for the

non-interacting scenario, while they haven’t provided many details of the split-step simulations for the non-linear interacting cases. I believe it is crucial for the authors to provide necessary descriptions of the simulation methods, as well as the simulation details, which can help the other researchers to reproduce the authors’ results and further extend the proposed methods to other similar cases.

b. Is there any specific reason for how the authors choose the initial positions for the singularities? Especially for the distance between multiple singularities in the homogeneous cases, the authors choose a = 0.5 for the 2-singularity case, but a = 1 for the 3-singularity case, is there any special consideration for these different choices?

Does the position parameter have any impact on the trajectory parameters other than

the merging time?

AUTHORS REPLY 2: We have included now a reference and an small description of the split-step method, together with the clarifications of the parameters used in the simulations (see second column, page 6). With regard to b, first we included a new figure 1 to clarify how the initial conditions look like and also Fig. 8 from new appendix A. We choose these initial positions of the singularities simply to find illustrative examples that exemplify clearly the method. For example that the merging time occurs close enough to be captured with the simulation, which is limited to a maximum time due to different aspects (e.g instability in the self-attractive case or the size of the box in any case – for the discussion of the new results on attractive interactions see ms and other referees reply).

COMMENT 3. Lacks in-depth discussion about the interesting cases:

a. In Fig. 1 (e), interestingly it shows the merging time of N = 20 doesn’t increase for g up to 0.38. I think it worth some comprehensive discussion on this case. Can the authors further elaborate on the reason behind this behavior?

AUTHORS REPLY 3a: We have included a more extensive discussion about this point in the new version of ms. We have repeated all calculations and also calculated with intermediated case, N=5 and N=10. We have found that in this latter case it increases slowly from $g=0$. We have also calculated how the merging point changes with g in the attractive case. This makes us conjecture that the change of merging point has a relationship with the behavior of the density – we have also included an appendix to illustrate the behavior of density. Nevertheless, we leave this for outlook and do not discuss in depth in ms. Note that (see figures 1 and 8) the larger the N the further away the ring of density is from the center of the system axis.

b. Other intriguing cases are the re-emergence and re-merging of the singularities in Fig. 2 (b), as well as the creation of two additional singularity pairs in Fig. 2 (c) and 4 (b). Can the author at least provide some high-level discussion on the mechanisms behind these phenomena?

AUTHORS REPLY 3b: We agree that this is a very interesting effect. Indeed this is a topic of our research even in the linear case for photonic systems when changes on discrete symmetry are involved and these phenomena occurs. We have checked that energy is conserved. Also notice that this emission is inside the vortex ring density. We have some conjecture about this but we have not a high level explanation. This is a topic for future research.

COMMENT 4. Equation issues & typos:

AUTHORS REPLY 4: we fixed all the typos and mistakes. We thank the referee for his/her acerful reading of ms.

Round 2

Reviewer 1 Report

I find the updated manuscript easier to follow than the previous version. However, the additions have made the manuscript somewhat bloated, and have introduced a number of new errors and inconsistencies which should be corrected before the work can be published. I suggest below some ways to clarify the presentation.

1. The introduction is now very long and lacks focus. I suggest the authors try to make it more concise by focusing on work that is directly relevant to this research.

2. I find the 3D "tube" plots of the vortex dynamics (Figs 3, 5 and 7) quite difficult to comprehend, especially when multiple vortex trajectories are present.
    (i) Could these plots be converted to the style of Fig 6? The trajectories are much clearer there.
    (ii) In any case, it would be useful to indicate in these plots where the edge of the system is, given that the wavefunction is expanding as a function of time (for g>=0). This could be done by finding where the modulus squared of the wavefunction drops to a few percent of the maximum.

3. Regarding the pair creation events seen in Figs 5 and 7, is it possible that these only occur in low density regions? Perhaps they should be considered "ghost" vortices that will only appear in regions of very low density? Addressing point 2(ii) above would make this clearer.

4. There seems to be a lot of repetition in the figures, so I suggest condensing the information in order to improve the presentation.
    (i) Figs 1 and 8 both show example initial conditions, so I do not think it is necessary to include both figures. Could one be removed? (See also point 5 below.)
    (ii) Figs 9, 10 and 11 all show very similar information, and take up a lot of space. Are all of these plots necessary to include? At the very least, could Figs 9 and 10 be combined into one 4x2 figure?
    (iii) Figs 12 and 13 add very little information to the trajectories already shown in Fig 6, so I suggest omitting both of these new figures (and the associated appendix). The observation that the vortex tends to the origin at late times in fig 6a can simply be stated in words.

5. With the new explanation of the initial conditions (and in particular the addition of Fig 8) I now understand the physical set up much more clearly. However, the authors rely heavily on Appendix A in the main text (referring to it multiple times, including spending two paragraphs describing its contents on page 7). If the reader must understand Appendix A in order to follow the main text, I suggest integrating the information into the main body of the manuscript. For example, Fig 8 could replace Fig 1, since both show the initial conditions, but Fig 8 also demonstrates how the normalisation changes the state.

6. In fig 4b, the blue and orange curves (corresponding to N=1 and N=20, respectively) look different to the previous version of the manuscript. They now cross at g~0.5, whereas in the previous version there was no crossing. What has changed in the calculation between the old and new version of this figure? Which is correct?

7. I noticed some inconsistencies around Figs 2, 3 and 4.
    (i) The curves in Fig 4c are described in the caption as "red, magenta, green". I think these should be "blue, magenta, red" for g=0,-0.25,-0.75, respectively.
    (ii) Fig 3e is described as having interaction strength g=-0.25, but it seems to correspond to the blue curve in Fig 4c, with a merging time of 3.4. As far as I understand, this is the merging time for g=0, N=1 (as pointed out in Fig 2). Could the authors clarify this apparent contradiction?
    (iii) In the description of Fig 4d, the orange curve is described as corresponding to N=20 in the caption, but N=10 in the main text (paragraph beginning "In Fig. 4(c) we plot..." on page 7). Which is correct?
    (iv) In the same paragraph, the N=5 and N=10 plots in Fig 4d are described as orange and green respectively, while the caption states the opposite. Could the authors correct this?

8. The authors introduce overbars in section 3 to denote dimensionless units, but then immediately drop the notation.
    (i) I suggest the authors simply state that they will measure time in units of 1/omega and distances in terms of the harmonic oscillator length.
    (ii) The overbar notation should also be removed from the axis labels in Fig 2 and 4b/d.

Minor comments:

1. In the final paragraph of page 7, the authors write "(as can be seen comparing with Fig. 9)". Should this sentence actually refer to Fig. 11?

2. In the Fig 1 caption, the positions appear to be quoted backwards compared to normal convention. For example, (0, +/-0.5) should be (+/-0.5, 0).

3. There are still a number of typos throughout the paper, so I suggest the authors do a more thorough proof-read.

Author Response

I find the updated manuscript easier to follow than the previous version. However, the additions have made the manuscript somewhat bloated and have introduced a number of new errors and inconsistencies which should be corrected before the work can be published. I suggest below some ways to clarify the presentation.

Authors reply: We thank the referee for their careful reading of the manuscript and insightful comments that helped us improve the content. We added all referees to the acknowledgements.

Comment 1. The introduction is now very long and lacks focus. I suggest the authors try to make it more concise by focusing on work that is directly relevant to this research.

Authors reply: We reduced the introduction by one paragraph, omitting the part that was unrelated to BECs but to photonics. We also removed the final part of paragraph one and some redundant text. We cut down references by 15. Introduction is a bit less than three columns, has a first part on generalities on vortices, one on  the mathematical definition, one describing what we do, one devoted to literature and context and a summary of paper. We judge it normal for a paper without any particular limit on length, and containing all what we find reasonable to contain. We agree that is reasonable to omit the part on photonics as we did.

Comment 2. I find the 3D "tube" plots of the vortex dynamics (Figs 3, 5 and 7) quite difficult to comprehend, especially when multiple vortex trajectories are present.

 (i) Could these plots be converted to the style of Fig 6? The trajectories are much clearer there.

 (ii) In any case, it would be useful to indicate in these plots where the edge of the system is, given that the wavefunction is expanding as a function of time (for g>=0). This could be done by finding where the modulus squared of the wavefunction drops to a few percent of the maximum.

Authors reply: With regard to comment 2i, we kept the tube figures as they are, because  they have shown useful with different audiences. For comment 2ii) please see reply to comment 4 ii).

Comment 3. Regarding the pair creation events seen in Figs 5 and 7, is it possible that these only occur in low density regions? Perhaps they should be considered "ghost" vortices that will only appear in regions of very low density? Addressing point 2(ii) above would make this clearer.

Authors reply: Please see reply to comment 4 ii)

Comment 4. There seems to be a lot of repetition in the figures, so I suggest condensing the information in order to improve the presentation.

 (i) Figs 1 and 8 both show example initial conditions, so I do not think it is necessary to include both figures. Could one be removed? (See also point 5 below.)

 (ii) Figs 9, 10 and 11 all show very similar information, and take up a lot of space. Are all of these plots necessary to include? At the very least, could Figs 9 and 10 be combined into one 4x2 figure?

 (iii) Figs 12 and 13 add very little information to the trajectories already shown in Fig 6, so I suggest omitting both of these new figures (and the associated appendix). The observation that the vortex tends to the origin at late times in fig 6a can simply be stated in words.

Authors reply:

  1. i) Fig. 1 has been substituted by Fig. 8, as per referee suggestion.
  2. ii) In the view of referee’s reviews and private communications with other colleagues we think that showing the density and phase is important to understand the method. We moved the figures to main text. We also included now an exemplary figure for the case of three singularities (new figure 8). In the new labelled figures 2, 3 and 4, and in figure 8 we included a dashed rectangle which shows the area plotted in Figs. 6 and 9. With regard to comment 2ii), we think this change helps visualize where the singularities locate. Other options, as the one suggested by referee, made the figures too complicated to understand in our opinion. With regard to comment 3, we added figure 8 and commented in text. The referee is right, in one sense, that the vortices appear in regions of low density. But comparing figure 8c and d with 8g and h shows that is not a trivial effect (the density is low in both cases, but outside the central ring). We think this simply deserves more attention, but this paper is devoted to introduce the method and compare with some numerical instances in the non-linear case. So it falls out of the scope of the paper.

iii) We still think that these figures show complementary information to Fig. 10 (label in new order), such as the frequency of oscillation. So we included them as small panels on top of the old figures (since as an inset they looked too small).

Comment 5. With the new explanation of the initial conditions (and in particular the addition of Fig 8) I now understand the physical set up much more clearly. However, the authors rely heavily on Appendix A in the main text (referring to it multiple times, including spending two paragraphs describing its contents on page 7). If the reader must understand Appendix A in order to follow the main text, I suggest integrating the information into the main body of the manuscript. For example, Fig 8 could replace Fig 1, since both show the initial conditions, but Fig 8 also demonstrates how the normalisation changes the state.

Authors reply: We removed both appendices.

Comment 6. In fig 4b, the blue and orange curves (corresponding to N=1 and N=20, respectively) look different to the previous version of the manuscript. They now cross at g~0.5, whereas in the previous version there was no crossing. What has changed in the calculation between the old and new version of this figure? Which is correct?

Authors reply: The current one is correct. In previous version the code for finding the merging point in the non-linear simulations was giving a shorter value for the case of N=20 because it was outputing the first relative minimum, but the singularites merged in several cases after some longer time.

Comment 7. I noticed some inconsistencies around Figs 2, 3 and 4.

(i) The curves in Fig 4c are described in the caption as "red, magenta, green". I think these should be "blue, magenta, red" for g=0,-0.25,-0.75, respectively.

Authors reply: The refreee is correct. We corrected this in the new version.

(ii) Fig 3e is described as having interaction strength g=-0.25, but it seems to correspond to the blue curve in Fig 4c, with a merging time of 3.4. As far as I understand, this is the merging time for g=0, N=1 (as pointed out in Fig 2). Could the authors clarify this apparent contradiction?

Authors reply: The referee is correct again. The figure labelled now as 6e is the linear case, with merging time at 3.4, and not the one for g=-0.25. We corrected this in the ms.

(iii) In the description of Fig 4d, the orange curve is described as corresponding to N=20 in the caption, but N=10 in the main text (paragraph beginning "In Fig. 4(c) we plot..." on page 7). Which is correct?

Authors reply:  The curve is for N=10. For N=20 the simulation instabilizes before merging small interactions. We corrected this in ms.

(iv) In the same paragraph, the N=5 and N=10 plots in Fig 4d are described as orange and green respectively, while the caption states the opposite. Could the authors correct this?

Authors reply: We corrected this. The figure for N=5 is green and the one for N=10 is the orange one.

Comment 8. The authors introduce overbars in section 3 to denote dimensionless units, but then immediately drop the notation.

 (i) I suggest the authors simply state that they will measure time in units of 1/omega and distances in terms of the harmonic oscillator length.

(ii) The overbar notation should also be removed from the axis labels in Fig 2 and 4b/d.

Authors reply: We removed the overbar notation in Eq. 19 and following ones and at the beginning of Sec. III.

Minor comments:

  1. In the final paragraph of page 7, the authors write "(as can be seen comparing with Fig. 9)". Should this sentence actually refer to Fig. 11?

Authors reply:  This is removed now, as the appendix is removed.

  1. In the Fig 1 caption, the positions appear to be quoted backwards compared to normal convention. For example, (0, +/-0.5) should be (+/-0.5, 0).

Authors reply:   We corrected this.

  1. There are still a number of typos throughout the paper, so I suggest the authors do a more thorough proof-read.

Authors reply: We fixed a number of typos.

Author Response

I thank the authors for their clarifications of the points raised in the previous report. I have some remaining concerns regarding the manuscript.

Authors reply: We thank the referee for his/her work, careful reading, and comments. We added all referees to acknowledgements, because they certainly helped improve the ms.

Main comments:

COMMENT 1 • From how I understand the normalization in Eq. (5), the authors choose to control atom number through the width of the initial Gaussian σ. This is unconventional in my view; normalization would usually be controlled through the prefactor A. As a result, the linear results in the paper depend on N . Can the authors justify this unconventional normalization? Perhaps the variation as a function of N would better be presented as variation as a function of σ? With an alternative normalization, the non-interacting results would be independent of N and the interacting results would depend on a single parameter gN .

Authors reply: We decided to keep N in the initial condition to keep full analogy with photonics, where it is the intensity of the laser beam. We do not find that keeping it in the initial condition is problematic for the linear case. The behavior depends on the shape of the initial condition, being it determined by sigma or A. We agree that comparing with the non-linear case would be easier, if one keeps a single parameter gN and aborbs N in A. One has to consider that here we intend to introduce the analytical method and compare with some instances of the non-linear case.

COMMENT 2. The authors clarify that their study focusses on singularities within the range of a vortex core. This is an interesting area of research. However, it is not clear how useful the results here are, since the initial condition isn’t justified in relation to known results on vortex core structure [see, for example, fig 1 of Phys. Rev. X 2, 041001 (2012)]. Is the Gaussian density profile justified physically? Does the choice of initial density profile have a major effect on the dynamics? The present understanding of a

vortex/anti-vortex collision is that it forms a Jones-Roberts soliton [J. Phys. A: Math. Gen. 15 2599 (1982); Phys. Rev. Lett. 119, 150403 (2017)]. The paper makes no attempt to link their results to this established understanding.

Authors reply: The paper has to be contextualized in the special issue "Computational Methods for Quantum Matter" from Condensed matter journal. We aimed at introducing a method from photonics to BECs, which worked for the linear case, and which one can compare with non-linear instances. We exemplified with some numerical examples. We left a lot of work to do, which we plan to do in the future as mentioned in outlook. The initial condition contains the vortices in the vortex core. And dynamics occurs mainly in vortex core. this we clarify in text. We included now the context suggested by the referee in the introduction. We emphasize it is not our goal to make a detailed study of vortices in vortex core. 

Other comments:

COMMENT 3 • The introduction is very long with a large amount of material not directly relevant to the manuscript. The purpose of the manuscript in the context of the literature is therefore not immediately clear.

Authors reply: We shortened the introduction by one paragraph, removing the part which was more related to photonics than to BECs. We also reduced some redundant text in the description of our goals and removed the final sentence of first paragraph. We cut down references by 15. Introduction is a bit less than three columns. It has a first part on generalities on vortices, one on  the mathematical definition, then already one a paragraph describing what we do. Next there is one paragraph devoted to literature and context and one devoted to a summary of paper. We judge it normal for a paper like this.  

COMMENT 4 • The extensive reference to Appendix A on page 7 is confusing. The authors should either include these results in the main manuscript or refer to the appendix in such a way that they do not disrupt the flow of the manuscript.

Authors reply:  We removed the appendices, and included the content in main text, Also we added Fig. 8.

COMMENT 5 • The use of overbars in the x and y coordinates in the paragraph below Fig. 2 (describing adimensionalization) is confusing, as this notation is previously used to denote complex conjugation. This notation is then dropped immediately. I think superfluous notation could be avoided, e.g.,  “time t will be measured in units of ω −1 and spatial coordinates (x, y) will be measured in units of p~/mω”.

Authors reply: We removed the overbar notation in Eq. 19 and following ones and at the beginning of Sec. III.

Reviewer 3 Report

The authors have properly responded to the comments left by the referee and made corresponding adjustments to the manuscript. With this, I think now the results presented in the paper are sound and the quality of the research meets the publication standards of Condensed Matter, therefore, I would recommend the acceptance for publication.

Author Response

The authors have properly responded to the comments left by the referee and made corresponding adjustments to the manuscript. With this, I think now the results presented in the paper are sound and the quality of the research meets the publication standards of Condensed Matter, therefore, I would recommend the acceptance for publication.

Authors reply: We thank the referee for his/her careful reading and comments. We added all referees to the acknowledgments because they really helped improve the manuscript.